# The kinetics of pre-mRNA splicing in the *Drosophila* genome and the influence of gene architecture

Athma A Pai[1], Telmo Henriques[2,3], Kayla McCue[4], Adam Burkholder[5], Karen Adelman[2,3], Christopher B Burge[1,4]*

[1]Departments of Biology and Biological Engineering, Massachusetts Institute of Technology, Cambridge, United States; [2]Epigenetics and Stem Cell Biology Laboratory, National Institute of Environmental Health Sciences, Research Triangle, United States; [3]Department of Biological Chemistry and Molecular Pharmacology, Harvard Medical School, Boston, United States; [4]Program in Computational and Systems Biology, Massachusetts Institute of Technology, Cambridge, United States; [5]Center for Integrative Bioinformatics, National Institute of Environmental Health Sciences, Research Triangle, United States

**Abstract** Production of most eukaryotic mRNAs requires splicing of introns from pre-mRNA. The splicing reaction requires definition of splice sites, which are initially recognized in either intron-spanning ('intron definition') or exon-spanning ('exon definition') pairs. To understand how exon and intron length and splice site recognition mode impact splicing, we measured splicing rates genome-wide in *Drosophila*, using metabolic labeling/RNA sequencing and new mathematical models to estimate rates. We found that the modal intron length range of 60–70 nt represents a local maximum of splicing rates, but that much longer exon-defined introns are spliced even faster and more accurately. We observed unexpectedly low variation in splicing rates across introns in the same gene, suggesting the presence of gene-level influences, and we identified multiple gene level variables associated with splicing rate. Together our data suggest that developmental and stress response genes may have preferentially evolved exon definition in order to enhance the rate or accuracy of splicing.

DOI: https://doi.org/10.7554/eLife.32537.001

*For correspondence:
cburge@mit.edu

## Introduction

Eukaryotic genes are generally composed of multiple exons with intervening introns that are spliced out to form mature RNA molecules. Despite the general conservation of this gene architecture, the number of introns per gene and relative sizes of exons and introns vary greatly across organisms (*Deutsch and Long, 1999*). For example, average intron lengths exceed 1 or 2 kilobases in most vertebrates compared to 50–500 nt in simpler eukaryotes, with somewhat less variability in exon lengths (*Deutsch and Long, 1999*). The presence and density of introns can influence gene expression levels (*Mascarenhas et al., 1990*; *Chung et al., 2006*; *Shaul, 2017*), while longer flanking intron lengths are associated with alternatively spliced exons (*Fox-Walsh et al., 2005*; *Gelfman et al., 2012*). It is not known whether pronounced differences in intron size across eukaryotic lineages has consequences for the efficiency of mRNA splicing.

The splicing of metazoan introns requires three primary sequence elements – the 5' splice site (at the exon/intron boundary), the 3' splice site and polypyrimidine tract (at and upstream of the intron/exon boundary), and the branch point sequence (BPS) – and the numerous (ribonucleo)proteins that make up the spliceosome. The earliest steps in splicing are the base pairing of the U1 and U2 small

ribonucleoproteins (snRNPs) to the 5' splice site and BPS, respectively. Progression toward splicing is thought to occur in one of two modes – either by 'intron definition,' in which the U1 and U2 snRNPs first interact across the intron; or 'exon definition,' in which U1 snRNP initially pairs with the upstream U2 snRNP across the exon, followed by rearrangement to form interactions with the downstream U2 snRNP across the intron (*Berget, 1995*). The pairing of U1 and U2 snRNPs requires bringing these complexes into close proximity, through either passive diffusion-based contact or co-localization regulated by other splicing factors (*De Conti et al., 2013*; *Hollander et al., 2016*). Subsequent steps of splicing are thought to proceed in a standard fashion regardless of the splice-site recognition mode, with the formation of an intronic lariat and cleavage at the 3' splice site to complete intron excision and joining together of the flanking exons.

Although intron splicing is likely the rate-limiting step in the production of many processed mRNAs, the precise mechanisms and consequences of differing splice-site recognition modes remains unclear. Previous observations suggest that the architecture of a gene – specifically the absolute or relative lengths of introns and exons (*Robberson et al., 1990*; *Sterner et al., 1996*; *Berget, 1995*) – is important for the selection of a splice-site recognition mode, with the shorter sized unit being favored for initial pairing of snRNPs (*Talerico and Berget, 1994*). In vertebrate genomes, where exon definition is thought to be the predominant recognition mode, the experimental expansion of exon lengths can prevent proper spliceosome formation (*Sterner et al., 1996*) except when the flanking introns are very short (*Chen and Chasin, 1994*). In yeast and *Drosophila*, the expansion of small introns can lead to intron retention and other splicing defects (*Guo et al., 1993*; *Talerico and Berget, 1994*). Some studies have suggested that recognition of introns and exons larger than 300–500 nt can be inefficient, unless the flanking units are much smaller (*Fox-Walsh et al., 2005*; *Sterner et al., 1996*; *Chen and Chasin, 1994*). There is also a minimum 5' splice site to branch point distance in each organism, which imposes a minimal intronic length, and splicing of very small exons is generally inefficient due to steric interference between bound factors (*Dominski and Kole, 1991*; *Black, 1991*; *Sterner et al., 1996*). For shorter intron-defined introns, intron length might limit the rate of diffusion-based contact between the U1 and U2 snRNPs (*Hollander et al., 2016*). However, for longer exon-defined introns which must undergo two recognition steps, this passive diffusion-based model would predict prohibitively slow splicing rates (*Hollander et al., 2016*).

We sought to understand the roles of exon and intron length in the kinetics of splicing to provide insight into the advantages and disadvantages of the longer introns present in more complex eukaryotes. *Drosophila* is an ideal organism to address this question, since it represents an invertebrate lineage known to have a mix of both intron-defined and exon-defined introns. Furthermore, *Drosophila melanogaster* has a broad distribution of intron lengths (*Lim and Burge, 2001*), with over a third of introns between 60 and 70 nt, and a tail of longer introns with lengths up to tens of kilobases (kb). Splicing kinetics has been studied for a limited number of introns using PCR-based or single-molecule imaging approaches. Splicing is thought to predominantly occur co-transcriptionally (*Singh and Padgett, 2009*; *Khodor et al., 2011*; *Schmidt et al., 2011*; *Brugiolo et al., 2013*), with measured rates for a handful of mammalian introns of 0.5–10 min (*Singh and Padgett, 2009*; *Schmidt et al., 2011*; *Martin et al., 2013*; *Rabani et al., 2014*) while yeast introns are often spliced within seconds after synthesis (*Eser et al., 2016*; *Oesterreich et al., 2016*). The kinetics of splicing can impact downstream gene expression pathways and overall gene expression levels. For instance, the splicing of U12-type introns (spliced by the minor spliceosome) can be rate-limiting, such that splicing – and the ultimate production of mature mRNAs – can be made more efficient by mutating U12-type splice sites to sites recognized by the major U2-type spliceosome in *Drosophila* cells (*Singh and Padgett, 2009*; *Patel et al., 2002*). Previous studies have suggested that alternatively spliced introns are spliced at different rates than constitutive introns (*Fong et al., 2014*; *Kwak et al., 2013*; *Jonkers et al., 2014*).

To globally assess the kinetics of RNA processing in *Drosophila melanogaster* and to address questions about the impacts of exon versus intron definition, we developed a new modeling approach and applied it to short time period metabolic labeling data. Together, these data enabled analysis of splicing rates from the introns of over 5000 *Drosophila* genes, revealing unexpected differences in the kinetics of intron versus exon definition, and identifying gene-specific features associated with differences in the speed of splicing.

## Results

mRNA splicing is a dynamic mechanism that can initiate immediately after an intron's transcription is completed. To measure the kinetics of mRNA splicing, it is necessary to capture nascent transcripts at short intervals after transcription, before introns have been completely spliced out. One effective experimental approach involves incorporating a metabolic label such as 4-thiouridine (4sU) to isolate RNA at short time points (within a few minutes) after transcription. The sequencing of nascent RNA over a time course of labeling periods is conceptually different from strategies used to estimate the extent of co-transcriptional splicing (e.g., Nascent-seq [*Khodor et al., 2011*, *2012*]) or to track the progression of PolII across genes (e.g., NET-seq [*Churchman and Weissman, 2011*; *Mayer et al., 2015*; *Nojima et al., 2015*]), focusing on time rather than location. This strategy has been used previously to assess relative splicing dynamics (*Windhager et al., 2012*), although previous studies lacked the power to quantitatively measure half-lives of intron excision across the genome (*Windhager et al., 2012*). To recover sufficient labeled RNA, it is generally necessary to incorporate 4sU for a period of minutes. This progressive labeling design results in the isolation of transcripts initiated prior to labeling but elongated during this period, as well as transcripts that initiated during the labeling period, yielding a range of times since completion of synthesis for any given intron (*Figure 1A*). We set out to develop an approach to derive the rate of intron splicing accounting for the distribution of times since synthesis in different transcripts resulting from progressive labeling.

### Comparison of approaches for assessing splicing rates

We compared three approaches for estimating splicing rates: (1) comparing intronic read density at long and short time points ('intron ratio'), as described previously in *Windhager et al. (2012)*; (2) modeling the rate of decrease in 'percent spliced in' (PSI or $\Psi$ for short, which estimates the proportion of unspliced introns) as a function of mean time since intron synthesis ('$\Psi$ decrease'); and (3) modeling the dynamics of splice junction appearance/unspliced junction disappearance, accounting for the effects of variable transcriptional completeness on splicing probability (*Figure 1B*). The intron ratio approach measures the ratio of intronic read density at a long labeling period (e.g. 60 min) to that at a short labeling period (e.g. 5 min); however, the use of only two time points may limit precision. Therefore, we developed the $\Psi$ decrease approach, which models splicing changes over a series of (three or more) time points, with $\Psi$ values calculated using the MISO software (*Katz et al., 2010*) as a measure of the proportion of transcripts that contain an intron at a given labeling period. Finally, to eliminate bias from reads deriving from incompletely synthesized introns (see Materials and methods), we developed a third approach that models the dynamics of the ratio of spliced exon-exon junction reads to unspliced intron-exon junction reads, accounting for the distribution of polymerase locations at the end of the labeling period.

To assess the performance of each approach and understand any systematic biases, we simulated reads from 4sU-labeled nascent sequencing data generated from a range of gene structures, gene expression levels and intron half-lives, at different simulated labeling durations (*Figure 1—figure supplement 1*). Overall, the junction dynamics approach was best able to accurately order introns by rate, and also accurately assign absolute rates to each intron in our simulations. Considering relative orderings, the $\Psi$ decrease and junction dynamics approaches both yielded rankings of half-lives very similar to those used in the simulation ($r_{Spearman}$ = 0.99 and 0.95, respectively), while the intron ratio approach correlated more weakly ($r_{Spearman}$ = 0.57; *Figure 1C*; *Figure 1—figure supplement 2A*), indicating much poorer agreement. This was also true when considering the relative fold differences in half-lives between pairs of introns (Materials and methods, *Figure 1E*; *Figure 1—figure supplement 2C*). Assessing absolute estimation of half-lives, we found that half-lives estimated by the junction dynamics approach differed by just 21% on average from the simulated values, while the $\Psi$ decrease approach tended to overestimate half-lives substantially (*Figure 1D*; *Figure 1—figure supplement 2B*). The junction dynamics method accurately recovered half-lives across the full range of simulated values ($r_{Pearson}$ = 0.95), with no detectable bias, while the $\Psi$ decrease method consistently overestimates absolute half-lives, with greater absolute deviation from true values at longer half-lives ($r_{Pearson}$ = 0.35; *Figure 1F*). Furthermore, the relative accuracy of the junction dynamics method does not significantly vary across the distribution of intron lengths (*Figure 1—figure supplement 2E*) or transcript length downstream of the 3' splice site (*Figure 1—figure*

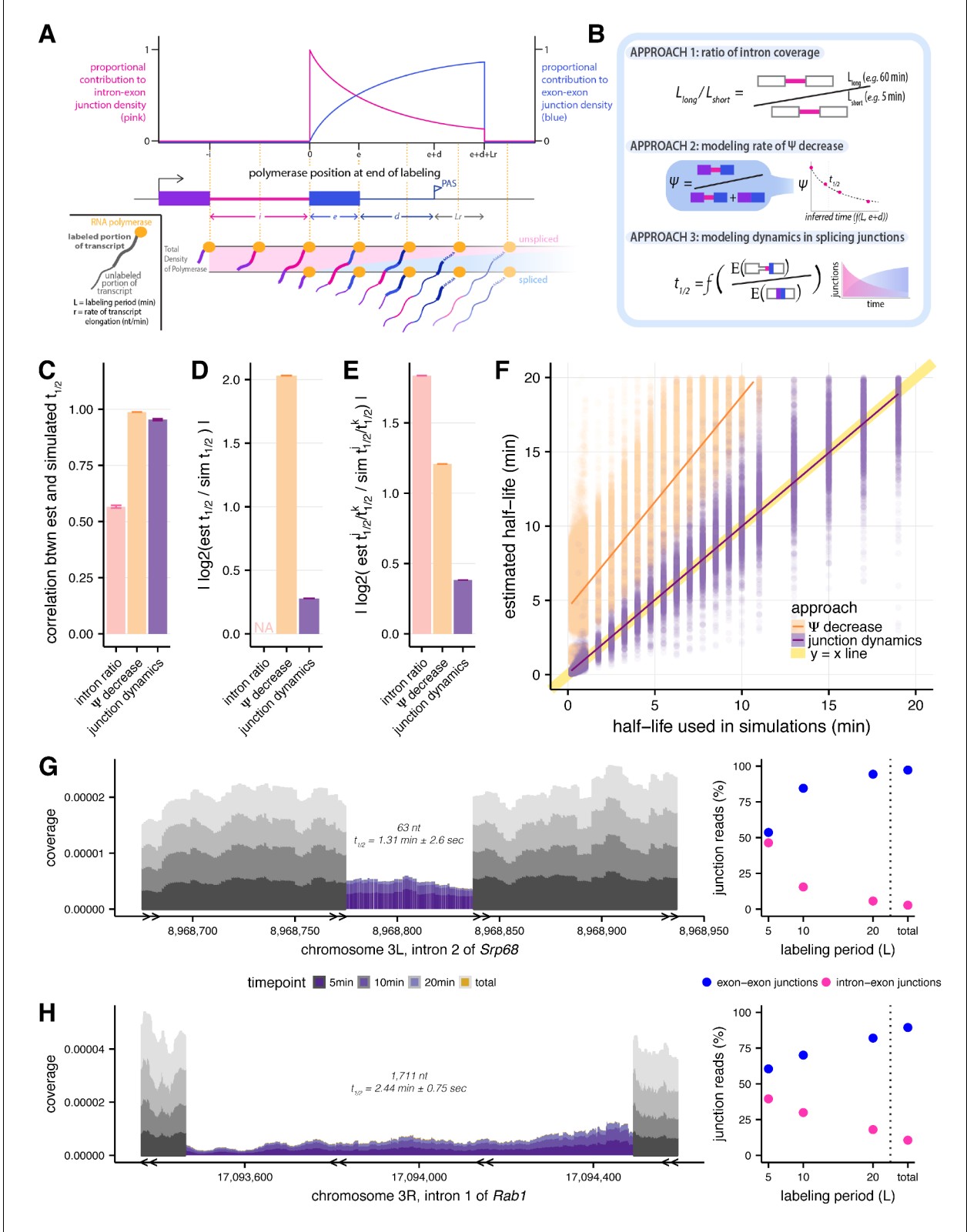

**Figure 1.** Estimating intron-specific splicing half-lives. (**A**) Progressive labeling with 4sU results in sampling of nascent RNA molecules from polymerase molecules distributed across a gene (*bottom*). The probability of sampling reads from unspliced or spliced transcripts, represented by intron-exon (pink) and exon-exon (blue) junction reads respectively, is dependent on the intron half-life and the location of the polymerase at the completion of the labeling period (*top*). (**B**) Schematics outlining the three approaches assessed for measuring rates of intron excision. (**C**) Mean Spearman correlations

*Figure 1 continued on next page*

*Figure 1 continued*

between simulated half-lives and estimated splicing rates from each of our three approaches (error bars are ± standard error). (D) Absolute percent error of our estimated splicing rates relative to simulated half-lives. Intron ratios are relative measures of half-lives, thus were not included in this comparison (error bars are ± standard error). (E) Relative absolute percent error of the estimated and simulated half-lives between two introns, allowing comparisons of metrics not expected to be drawn from the same distribution (error bars are ± standard error). (F) Estimated half-lives from Ψ decrease and junction dyanmics approaches (*x-axis*) versus to the half-lives used to simulate read data (*y-axis*). Yellow line indicates y = x line of perfect correlation. (G and H) Nascent RNA coverage across the second intron of *Srp68* (G) and the first intron of *Rab1* (H). Colors represent time points, with 5 min after 4sU labeling (*darkest shade*), through 10 min, 20 min, and total RNA sample (*lightest shade*). Right panels show the proportions of intron-exon (*pink*) and exon-exon (*blue*) junction reads out of all 3′ junction reads in each labeling period.

DOI: https://doi.org/10.7554/eLife.32537.002

The following figure supplements are available for figure 1:

**Figure supplement 1.** Simulating 4sU-seq reads.
DOI: https://doi.org/10.7554/eLife.32537.003
**Figure supplement 2.** Assessing approaches for estimating intron-specific splicing half-lives.
DOI: https://doi.org/10.7554/eLife.32537.004
**Figure supplement 3.** Applying the junction dynamics approach to estimate intron-specific half-lives in *Drosophila* cells.
DOI: https://doi.org/10.7554/eLife.32537.005

*supplement 2F*). The advantages presented by the junction dynamics approach come at the expense of somewhat reduced power to measure splicing rates in lower-expressed genes (*Figure 1—figure supplement 2D*), where numbers of captured splice junction reads may be too low for analysis. Taken together, the results of our simulations clearly indicate that the junction dynamics method provides the best overall combination of absolute and relative accuracy in inference of splicing half-lives from progressive labeling 4sU-seq data.

## Measuring rates of splicing for 25,000 *Drosophila* introns

We applied the junction dynamics method to a time course of 4sU-seq data generated from *Drosophila melanogaster* S2 cells. Our experimental approach involved 5, 10, and 20 min labeling with 4-thiouridine (4sU), followed by RNA sequencing of three replicates per labeling period. These data were complemented by steady state RNA-seq data representing predominantly mature RNA (Materials and methods). The relative ratio of intron-exon junction reads to exon-exon junction reads was highest at the 5-min labeling period and decreased rapidly to low levels at longer times (*Figure 1—figure supplement 3A*). For the 1000 most highly expressed genes, intron half-lives derived from individual replicates had lower coefficients of variation than comparisons across labeling periods (mean $r_{Pearson}$ across replicates = 0.67; *Figure 1—figure supplement 3B*). Therefore, we increased our power to measure rates in lower-expressed genes by pooling the reads from the three replicates of each labeling period.

Applying our junction dynamics method, as described above, we obtained intron half-lives with associated confidence intervals and goodness-of-fit statistics for 25,576 constitutive introns in 5608 *Drosophila* genes with sufficient expression in S2 cells (Materials and methods, representative examples in *Figure 1G–H*; *Figure 1—figure supplement 3C–D*; *Supplementary file 1*). The median intron half-life ($t_{1/2}$) was 2.0 min, with most half-lives estimated here lying between those previously estimated in yeast (seconds) and mammalian cells (30 s to 10 min) using qRT-PCR or imaging approaches (*Oesterreich et al., 2016*; *Eser et al., 2016*; *Singh and Padgett, 2009*; *Martin et al., 2013*; *Coulon et al., 2014*). For typical *Drosophila* genes of 3–9 kb in length, the estimated time of transcription is about 2–6 min (*Ardehali and Lis, 2009*). Thus, our estimated median intron half-life of ~2 min is consistent with splicing occurring co-transcriptionally for the majority of fly introns (*Khodor et al., 2011*; *Brugiolo et al., 2013*; *Braunschweig et al., 2013*). Our approach assumes an average transcription rate across the gene body of 1.5 kb/min (*Ardehali and Lis, 2009*; *Garcia et al., 2013*). Estimated half-lives are robust with respect to this parameter, varying by only ~10% on average for transcription rates between 1 and 3 kb/min (*Figure 1—figure supplement 3E*). Our approach also assumes that the rate of transcription elongation is uniform across the gene, in the absence of genome-wide local elongation rate data; gross deviation from this assumption could influence variability in splicing rates between genes or between introns in different positions along a gene.

## Introns 60–70 nt in length are spliced rapidly

Intron length may influence splicing kinetics (*Proudfoot, 2003*; *Hicks et al., 2010*; *Windhager et al., 2012*; *Khodor et al., 2011*). The distribution of lengths of *Drosophila* introns has a sharp peak at 60–70 nt, with more than half of introns between 40 and 80 nt in length, and the remainder distributed over a broad range extending beyond tens of kilobases (*Lim and Burge, 2001*) (*Figure 2—figure supplement 1A*). While there was an overall weak positive relationship between intron lengths and estimated splicing half-lives across the entire distribution ($r_{Spearman}$ = 0.1, p-value<$2.2 \times 10^{-16}$), a notable decrease in half-life occurred near the short end of the intron length distribution (*Figure 2A*). Examining the rates for different bins of intron length in this vicinity, we found that introns with lengths in the range 60–70 nt were spliced most rapidly (median $t_{1/2}$=1.7 min, *Figure 2B*). Median $t_{1/2}$ increased steadily to 2.2 min for introns > 80 nt, and was also substantially higher for 'ultra-short' introns between 40–50 nt (median $t_{1/2}$ of 2.7 min). These ultra-short introns had substantially weaker 3′ splice site scores than other introns (*Farlow et al., 2012*) (*Figure 2—figure supplement 1B*), and 60–70 nt introns with similarly weak 3′ splice sites were spliced at similar rates (*Figure 2—figure supplement 1C*), suggesting that 3′ splice site weakness is a cause of slower splicing.

The 3′ splice site is generally considered to consist of a four nt core motif (YAG/G) preceded by a ~ 10–15 nt polypyrimidine track (PPT) (*Reed, 1989*) that is often absent in shorter *Drosophila* introns (*Mount et al., 1992*; *Kennedy and Berget, 1997*). We hypothesized that the weaker 3′ splice sites and slower splicing of ultra-short *Drosophila* introns might result from constraints on branch point sequence (BPS) position, since minimum 5′ss-BPS and BPS-3′ss distances are required for splicing (*Mount et al., 1992*; *Wieringa et al., 1984*). In flies, these minima appear to be ~35 nt and ~15 nt, respectively (*Mount et al., 1992*). Thus, the BPS location in a 45 nt intron is constrained to fall between positions −15 and −10 relative to the 3′ss, a position that leaves little room for a PPT, likely explaining the weaker 3′ss scores in this class. To explore this possibility, we scanned for a BPS motif in introns from the different length classes (as in [*Lim and Burge, 2001*]). In each length class, we identified a plausible BPS motif with consensus CTAAT (*Mount et al., 1992*) (*Figure 2C*). However, the mean location of this motif was much closer to the 3′ss in ultra-short introns (−14) than in the longer intron classes, where the mean location was between −23 and −26. This observation suggests that ultra-short introns are constrained in their ability to splice efficiently by their suboptimal BPS locations, which may simply prevent a sufficiently long PPT for efficient splicing, or may result in steric interference between dU2AF[50] bound at the PPT and U2 snRNP bound at the BPS (*Black, 1991*; *De Conti et al., 2013*). These observations, in conjunction with evidence that natural selection favors short intron lengths in *Drosophila* (*Carvalho and Clark, 1999*; *Parsch, 2003*; *Parsch et al., 2010*), suggests that selection on intron lengths for efficient splicing contributes to the observed peak of intron size near ~65 nt.

## Alternative introns are spliced more slowly than constitutive introns

Studies mostly in mammalian systems have observed that alternative introns may have different kinetic parameters than constitutive introns (*Pandya-Jones and Black, 2009*; *Fong et al., 2014*; *Kwak et al., 2013*; *Jonkers et al., 2014*). To explore whether similar differences exist in *Drosophila*, we analyzed alternatively retained introns (RI), using introns that are annotated as retained in other fly cell types or tissues but are constitutively spliced out in S2 cells. We observed that such RIs have significantly slower splicing half-lives than constitutive introns (CI), independent of intron length, as do introns that flank exons alternatively skipped in other cells/tissues (SEflanking) (all Mann-Whitney p<0.01 for all comparisons between CI and RI and between CI and SEflanking; *Figure 2D*). This observation suggests that the capacity for splicing regulation may impose limits on the rates of intron splicing.

## Exon definition is associated with faster and more accurate splicing

One hallmark of the *Drosophila melanogaster* splicing program is the common use of both intron definition and exon definition modes of initial splice site recognition (*Berget, 1995*). However, it is not known whether this difference in mode has any functional consequences for the rate or accuracy of splicing. Detailed biochemical studies of single *Drosophila* introns have observed that exon and intron length play important roles in governing recognition mode (*Berget, 1995*; *De Conti et al.,*

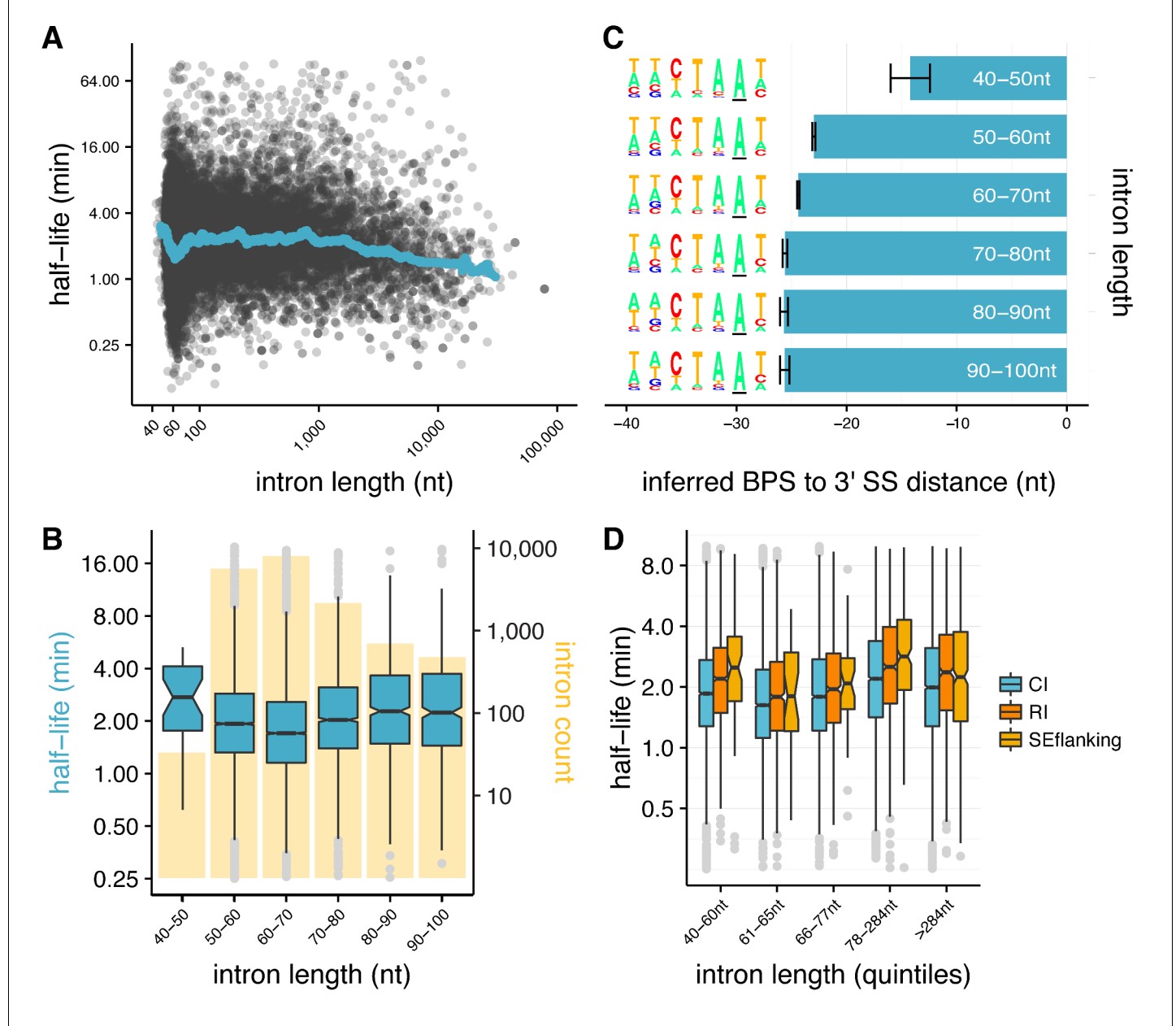

**Figure 2.** Splicing efficiency is variable across classes of intron length and regulatory potential. (**A**) Running median of splicing half-lives across distribution of intron lengths. Median is computed in sliding bins of 50 introns. (**B**) Splicing half-lives across bins of 10 nt intron lengths. Background bars display number of *Drosophila* introns in each bin. (**C**) Mean position of branchpoint 'A' in the strongest branchpoint motif in each bin of intron length (error bars are ± standard error). Motif logos (*left*) were created using branchpoint motif across all introns in the intron length bin. (**D**) Splicing half-lives in different categories of intron regulation, where constitutive introns (*blue*) are spliced out faster than either annotated regulated retained introns (RI, *orange*) or introns flanking annotated alternative exons (SEflanking, *yellow*), for cases where the RI is fully spliced and the SE is fully included in S2 cells.

DOI: https://doi.org/10.7554/eLife.32537.006

The following figure supplement is available for figure 2:

**Figure supplement 1.** Properties of splicing efficiency across varying intron lengths.

DOI: https://doi.org/10.7554/eLife.32537.007

*2013*). A general consensus has emerged that shorter introns and/or longer exons favor intron definition, and shorter exons and longer introns favor exon definition. However, there is less consensus about the relative importance of exon versus intron lengths (*Robberson et al., 1990*; *Talerico and Berget, 1994*; *Sterner et al., 1996*; *Chen and Chasin, 1994*; *De Conti et al., 2013*; *Fox-Walsh et al., 2005*). Here, we used the Ratio of Intron length to Mean flanking Exon length (RIME) to assess the contributions of local intron/exon lengths to splicing mode (*Figure 3A*; *Figure 3—figure supplement 1A*). Based on the consensus noted above, introns that are shorter than the mean length of their flanking exons (RIME <1) are more likely to undergo intron definition, while introns with RIME values > 1 may favor exon definition.

To understand the relationship between intron and flanking exon lengths and splicing half-lives, we assessed splicing rates across bins of joint intron-exon length and RIME values (Materials and methods; *Figure 3B*; *Figure 3—figure supplement 1A–B*). This representation allows us to observe the effects of RIME on splicing rates while controlling for the uneven distribution of intron and exon lengths in different RIME value bins (*Figure 3—figure supplement 1C*). Consistent with our observations above, introns between 60 and 70 nt contribute to a prominent enrichment of fast splicing at lower RIME values (vertical 'stripe' of yellow bins near left side of *Figure 3C*; *Figure 3—figure supplement 1D–E*), with introns < 60 nt visible as a narrower purple stripe of bins with lowest RIME values at far left of *Figure 3C*(*Figure 3—figure supplement 1D*). Also notable was a horizontal yellow stripe representing long introns with the highest RIME values, while introns with RIME near one tended to have longer splicing half-lives (*Figure 3D–E*; *Figure 3—figure supplement 1D*). This pattern is consistent with the idea that length combinations more predictive of exon definition (RIME > 1) or intron definition (RIME < 1) were spliced more rapidly, while intermediate combinations with ambiguous splicing mode were associated with slower splicing (see Discussion). For introns likely to undergo intron definition (RIME <<1), the longest intron lengths (>99 nt) were associated with the slowest splicing. In contrast, for introns likely to undergo exon definition, the longest intron lengths (>2944 nt) were associated with the fastest splicing rates (*Figure 3F*). This surprising observation suggests that exon and intron definition modes may have fundamentally different kinetics, perhaps with different rate-limiting steps (see Discussion). Overall, intron length showed a stronger correlation with splicing half-lives than the mean length of the flanking exons (*Figure 3—figure supplement 1F*).

Efficient recognition of exons may also be promoted by binding of splicing regulatory factors. For example, binding of SR proteins to exons may aid in exon recognition (*Berget, 1995*; *De Conti et al., 2013*). To explore this potential contribution, we identified hexanucleotides (6mers) enriched in introns and flanking exons of introns with high and low RIME values. Strikingly, putatively exon-defined introns (high RIME) had an abundance of significantly enriched 6mers in the flanking exons but no significant 6mers in intronic regions (*Figure 3G*; *Figure 3—figure supplement 2A*). In contrast, low RIME introns had many significant 6mers in intronic regions, and relatively few in the flanking exons. The overall asymmetry in 6mer enrichment supports a role for region-specific sequence features in splice-site recognition modes.

Many of the 6mers enriched in specific regions match binding motifs for known splicing regulatory factors. For example, AACAAC, CAACAA, and ACAACA, among the top enriched 6mers in both upstream and downstream exons flanking exon-defined introns, were among the five most enriched 6mers in exons flanking alternative exons regulated in S2 cells by the *Drosophila* splicing factor Pasilla (PS), a homolog of mammalian Nova family splicing factors which bind a distinct but related motif, YCAY (*Brooks et al., 2011*). The most enriched 6mer in downstream exons, GCAGCA, is a known mammalian exonic splicing enhancer (*Fairbrother et al., 2002*) and a binding motif for the SR proteins SRSF8 and SRSF10 (Dominguez *et al.*). The shorter splicing half-lives of high RIME ('exon-defined') introns might reflect more efficient recognition of their flanking exons. Indeed, longer exon-defined introns – which are spliced more quickly – are preceded by upstream exons with a higher density of significant 6mers (in aggregate) than shorter exon-defined introns (*Figure 3—figure supplement 2B*).

Different modes of initial splice site pairing might also induce different rates of splicing errors. As a simple measure of potential splicing errors, we computed the fraction of reads that spanned 'non-canonical' splice junctions, (i.e. pairs of intron terminal dinucleotides other than the three canonical pairs 'GT-AG,' 'GC-AG,' and 'AT-AC' that account for ~99.9% of all known fly introns). Despite being longer, exon-defined introns (high RIME) had the lowest mean frequency of non-canonical junction

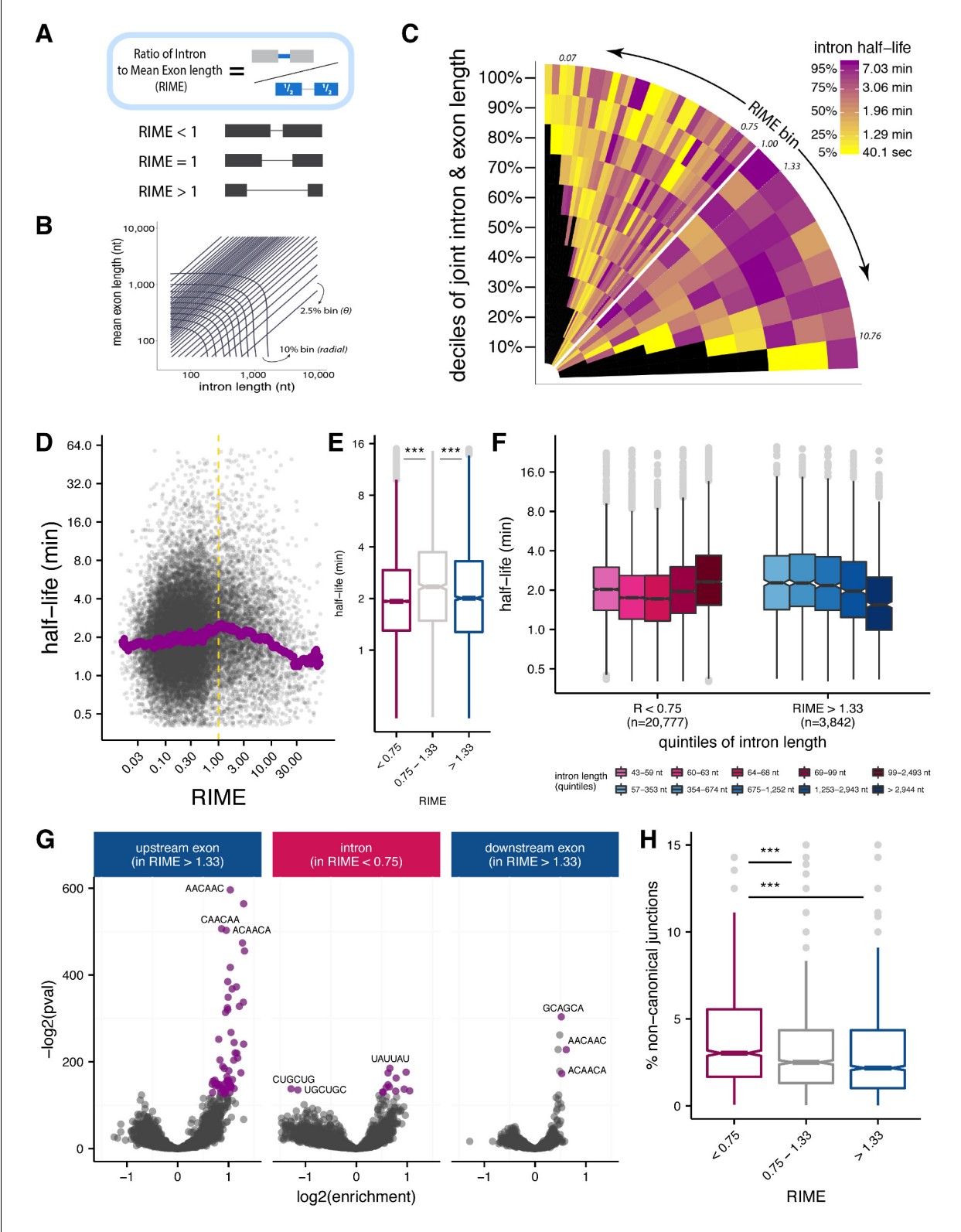

**Figure 3.** Splice site recognition mode influences the efficiency of splicing. (**A**) The Ratio of Intron to Mean Exon (RIME) metric is defined as the ratio of a given intron length to the mean length of the exons flanking that intron. (**B**) Schematic of binning to capture intron length (*x-axis*, nt), mean flanking exon length (*y-axis*, nt), and RIME values in a single plot. Radial bins capture both intron and mean exon length (10% bins), while diagonal bins (2.5% bins, *θ axis*) capture the RIME distribution. (**C**) Distribution of mean splicing half-lives across bins of RIME (*θ axis*) and deciles of joint intron and

*Figure 3 continued on next page*

*Figure 3 continued*

exon lengths (*r axis*). Yellow represents short mean half-lives and dark purple represents longer mean half-lives. (**D**) Running median of splicing half-lives across distribution of RIME values. Median is computed in sliding bins of 200 introns. (**E**) Distribution of splicing half-lives for introns with RIME <0.75 (*pink*), 0.75 < RIME < 1.33 (*grey*), and RIME >1.33 (*blue*). (**F**) Splicing half-lives across quintiles of intron length in each RIME class (RIME <0.75 in shades of pink on left and RIME >1.33 in shades of blue on right). (**G**) Enrichment of *6mers* in exons upstream of introns with RIME >1.33 (*left*), intronic regions of introns with RIME <0.75 (*middle*), and exon downstream of introns with RIME >1.33 (*right*). Significant *6mers* are in purple (Benjamini-Hochberg corrected p-value<$10^{-30}$). (**H**) Splicing accuracy measured by percentage of non-canonical unannotated reads for introns with RIME <0.75 (*pink*), 0.75 < RIME < 1.33 (*grey*), and RIME >1.33 (*blue*).

DOI: https://doi.org/10.7554/eLife.32537.008

The following figure supplements are available for figure 3:

**Figure supplement 1.** Jointly evaluating effects of intron and exon length on splicing half-lives.

DOI: https://doi.org/10.7554/eLife.32537.009

**Figure supplement 2.** Enrichment of sequence elements in and around introns with variable RIME values and lengths.

DOI: https://doi.org/10.7554/eLife.32537.010

reads, while intron-defined introns (low RIME) had highest frequency of non-canonical splice site usage (*Figure 3H*). This observation suggests that the initial recognition of splice sites across exon units involved in exon definition may improve the accuracy of splice site choice.

## Consistent splicing rates across introns in a gene

To explore what other variables might influence the efficiency of splicing of introns, we used a multiple linear regression model to estimate the contribution of various candidate features to splicing half-lives (Materials and methods). We separately fit this model to RIME-inferred intron-defined introns (*n* = 12,990) and exon-defined introns (*n* = 1,873), and restricted our analysis within each group to non-first introns for reasons discussed below (Materials and methods; *Figure 4A* and *Figure 4—figure supplement 1A–B*). Overall, these variables accounted for 19.9% and 16% of variance in splicing half-lives for intron- and exon-defined introns, respectively. Intron length was one of the strongest contributors for both sets of introns, contributing positively to splicing rate for the exon-defined class and negatively for the intron-defined class, consistent with results shown in *Figure 3E*. Exon lengths accounted for less variation in half-lives than intron lengths, although the direction was opposite to that of intron lengths for each splicing mode. As expected, increased strength of both the 5' and 3' splice sites was associated with shorter intron half-lives (*Hicks et al., 2010*). However, 5' splice site strength appeared more important for intron-defined introns, while 3' splice site strength had a greater impact for exon-defined introns, consistent with models of splice site recognition (*De Conti et al., 2013*; *Hollander et al., 2016*). A + U content of the intron was also associated with shorter intron half-lives, especially in the 3' region of the intron (excluding the 3' splice site). Introns are U-rich in many metazoans and plants, and many proteins of the heterogeneous nuclear ribonucleoprotein (hnRNP) class bind A + U rich motifs in introns and participate in splicing (*Goodall and Filipowicz, 1989*; *Lorković et al., 2000*). As the BPS motif consensus is also A + U rich, increased BPS strength might also contribute to this observation.

One of the features most strongly associated with faster splicing in both splice site recognition modes was gene expression level, a property of the gene rather than of the intron. This relationship is consistent with our observation that more rapidly spliced introns, particularly exon-defined introns, tend to occur in higher expressed genes (*Figure 4—figure supplement 1C*). Greater first intron length was also associated with faster splicing of other introns for the intron definition class. The importance assigned to gene expression and first intron length, both gene-specific rather than intron-specific properties, raised the question of the relationship between splicing rates of different introns in the same gene. We observed that splicing half-lives of introns drawn from the same gene tend to be more similar to each other (than those of randomly sampled introns, for example, comparing standard deviations (Mann-Whitney comparison of SDs, p<$2.2 \times 10^{-16}$; *Figure 4B*). Randomly varying the rate of elongation across introns within a gene yielded the same result, suggesting that this result is not dependent on our assumption of a uniform transcription rate across the gene (*Figure 4—figure supplement 1D*). This relationship remained when considering the coefficient of variation rather than the standard deviation (*Figure 4—figure supplement 1E*). Such a relationship might result indirectly from differing levels of selection for efficient splicing experienced by different genes

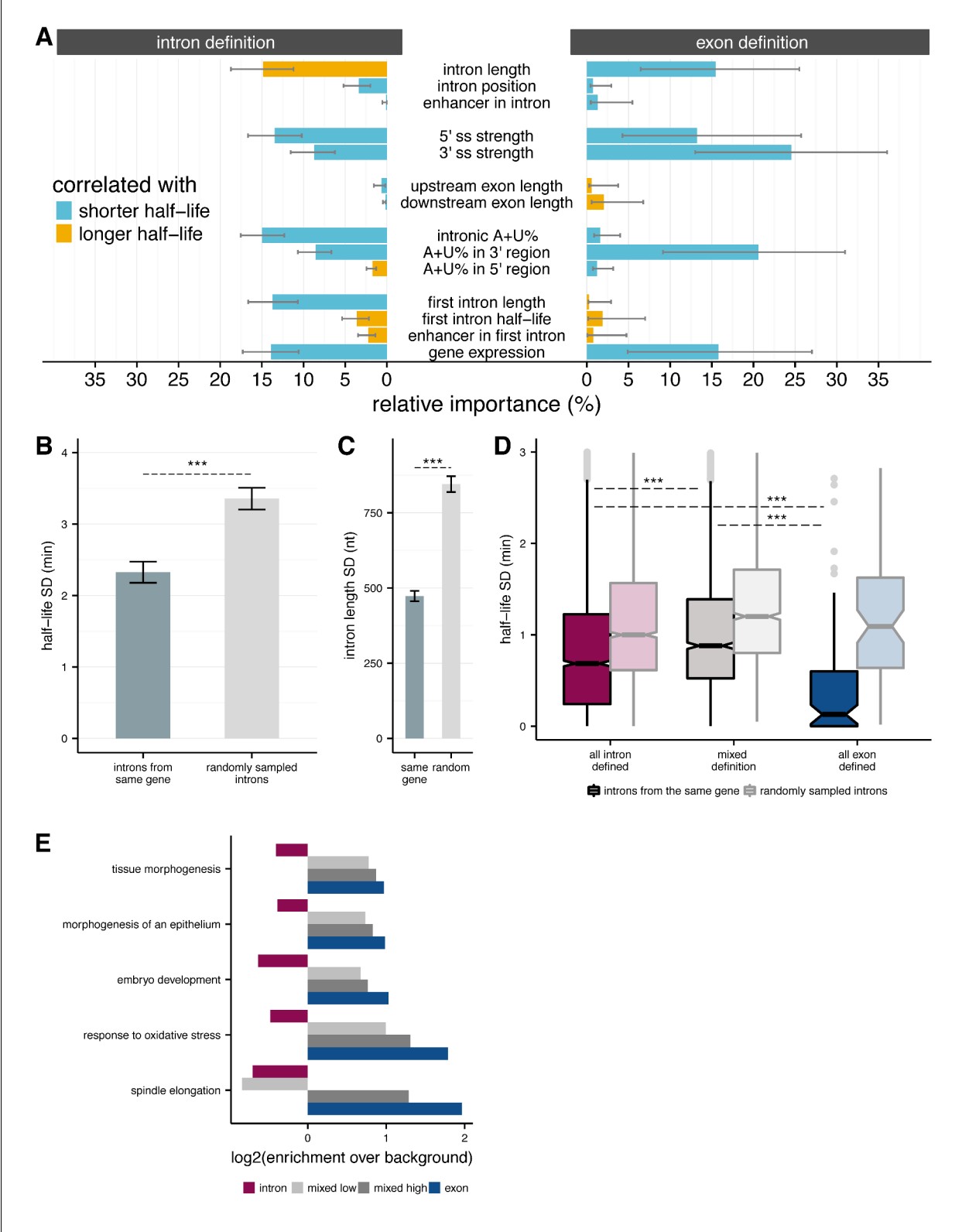

**Figure 4.** Splicing efficiency across introns within a gene. (**A**) Relative importance of variables influencing variance in splicing half-lives in intron-defined (*left*) and exon-defined (*right*) introns, using a multiple linear-regression to account for variance in half-lives for non-first introns. (**B**) Mean variance of splicing half-lives across introns within a gene relative to randomly sampled introns (chosen to match the distribution of lengths within actual genes; error bars are ± standard error). (**C**) Average standard deviation of intron lengths across introns within a gene relative to randomly sampled introns

*Figure 4 continued on next page*

*Figure 4 continued*

(error bars are ± standard error). (D) Average standard deviation of splicing half-lives across introns within genes with mostly intron-defined introns (*left*), mostly exon-defined defined introns (*right*), and a mixture of definition classes (*middle*), relative to randomly sampled introns within each category of genes (*lighter colors*). (E) Enrichment of genes (*x-axis; log2*) within Gene Ontology categories that are significantly over-represented among exon-defined genes (*y-axis*) for classes of genes with increasing proportions of exon-defined genes: all intron-defined (*pink*), mixed-definition (*grey*), and exon-defined (*blue*).

DOI: https://doi.org/10.7554/eLife.32537.011

The following figure supplements are available for figure 4:

**Figure supplement 1.** Intron- and gene-specific variables contributing to variability in splicing half-lives.
DOI: https://doi.org/10.7554/eLife.32537.012

**Figure supplement 2.** First-intron length and splicing efficiency.
DOI: https://doi.org/10.7554/eLife.32537.013

or classes of genes, or directly from gene-level features that impact the splicing efficiency of all introns in a gene.

Introns from the same gene are also more likely to have similar lengths than randomly sampled introns (Mann-Whitney comparison of SDs, $p < 2.2 \times 10^{-16}$, **Figure 4C**), suggesting that there might be an advantage to maintaining a consistent splice site recognition mode across introns within a gene. We classified exon/intron definition preferences at the gene level by dividing genes into three classes: (1) all or mostly intron-defined, with $\geq 2/3$ intron-defined (n = 4,463); (2) all or mostly exon-defined ($\geq 2/3$ exon-defined) (n = 308); and (3) the remaining 'mixed genes' with substantial proportions of introns from both recognition modes (n = 837; Materials and methods). We observed that splicing half-lives of introns drawn from genes with consistent splice site pairing preferences (classes 1 and 2) tend to be more similar to each other than introns from mixed definition genes. Notably, splicing half-lives across introns from genes with mostly exon-defined introns were less than half as variable than half-lives across introns from intron-defined genes (**Figure 4D**).

## First introns are slowly spliced and may influence rates of downstream introns

It was intriguing that the presence of a long first intron was associated with faster splicing of subsequent non-first introns within a gene, particularly for genes with predominantly shorter intron-defined genes. In *Drosophila*, first introns are more than twice as long on average as other introns (**Figure 4—figure supplement 2A**) and take at least 50% longer to splice than second, third, fourth, or other intron positions. In metazoans, transcriptional enhancers are often located in the first introns of genes (**Kharchenko et al., 2011**; **Arnold et al., 2013**), and we confirmed this trend in genes expressed in S2 cells (**Figure 4—figure supplement 2B**). Thus, the frequent presence of transcriptional enhancers may contribute to increased lengths of first introns, and could possibly impact the kinetics of splicing of first or non-first introns. Presence of a transcriptional enhancer in any intron (first and non-first) was associated with slower splicing of intron-defined introns and faster splicing of exon-defined introns (**Figure 4—figure supplement 2C**), but this association was not significant for introns from either splice site recognition mode in the regression model of splicing rates including other variables described above, so was not further explored here. Splicing half-lives or lengths of the last (3' most) intron in the gene were not significantly different than those of internal introns.

We observed a weak but significant negative correlation between the length of the first intron in a gene and the median half-lives across downstream introns ($r_{spearman} = -0.06$; p=0.00034; **Figure 4—figure supplement 2D**). This relationship remained when controlling for variability in the lengths of non-first introns by restricting the analysis to genes whose non-first introns were all intron-defined with lengths between 60 and 70 nt (**Figure 4—figure supplement 2E**). These observations suggest that the architecture of the 5' end of the gene, notably the length of the 5'-most intron, impacts the splicing efficiency of the remaining introns. These observations, together with our results in **Figure 4A and B**, suggest that various properties of the gene – including its overall exon-intron architecture – impact the splicing rate of all introns, rather than each intron's splicing rate being independently determined.

## Discussion

The rates of particular steps in gene expression can provide clues to mechanism, and in some cases selection may act on these rates. Since all the introns in a transcript must be removed to produce a mature mRNA that can be exported and translated, natural selection (if present) may act primarily on the total elapsed time to splice all introns rather than on the splicing rates of individual introns. We observed above that longer first introns were associated with faster splicing of downstream introns. Longer first introns might allow additional time for the recruitment of factors that promote downstream splicing efficiency, consistent with the observation that transcription elongation is slow before reaching the second exon, at least in mammalian cells (*Jonkers and Lis, 2015*). Under this model, longer first introns should be favored in genes with larger numbers of introns, where the splicing time of the first intron is less likely to be rate-limiting, because of the longer time required for the polymerase to reach the 3′ end of the transcript and the greater likelihood that another intron's splicing is rate-limiting for production of mature mRNA. Consistent with this expectation, we observed a pronounced trend where genes with larger numbers of introns have longer first introns (*Figure 4—figure supplement 2F*).

There is a well-established mutational bias toward short deletions (over short insertions) in the *Drosophila* lineage (*Petrov et al., 2000*). This bias may explain the predominance of short introns in the *D. melanogaster* genome (*Comeron and Kreitman, 2000*; *Parsch, 2003*; *Lim and Burge, 2001*). Our observation that the 60–70 nt length range – where there is a sharp peak in the genome-wide intron length distribution – represents a local maximum in intron splicing rates, suggests that this peak may have been shaped by selection for efficient splicing. At a length of 60–70 nt (or shorter), the 5′ and 3′ splice sites and BPS occupy a large fraction of the intron. Thus, length-changing mutations may often disrupt recognition of core motifs in this class, contributing to maintenance of intron length. Absent data comparing splicing dynamics across species, it is difficult to estimate selective pressures on splicing half-lives and intron lengths. Our observation that gene architectures characteristic of exon definition are associated with not only faster but also more accurate splicing prompts us to speculate that selection for splicing speed and/or accuracy may play a role in maintenance of long introns in genes with particular biological functions (discussed below), despite the expected mutational pressure toward shortening. It is also possible that variable mutational biases across the *Drosophila* genome explain the presence of long introns in a subset of genes.

It is less clear why exon definition should be associated with increased speed and accuracy. Perhaps, the increased time required to transcribe the longer introns characteristic of exon-defined transcripts affords more time for recognition (and perhaps proofreading) of exon definition complexes on upstream exons. It has also been proposed that efficient juxtaposition of exons is favored by tethering of the upstream exon definition complex to the elongating polymerase during transcription of the subsequent exon (*Hollander et al., 2016*). Splicing factors of the SR protein family or other families may also play a role in accelerating exon definition, given our observations that particular sequence motifs are enriched in exons associated with exon definition, and previous observations that SR proteins can enhance splicing rates in mammalian systems (*Long and Caceres, 2009*; *Zhou et al., 2013*).

We also observed that genes with mixed modes of initial splice site recognition had less consistent splicing rates of their introns than genes with predominantly intron or exon definition. If initial pairing of U1 and U2 snRNPs occurs across an intron, the U2 complex bound at the 3′ end of the intron might have reduced availability to pair with a U1 snRNP bound to the downstream 5′ splice site, inhibiting exon definition downstream, and conversely (*Berget, 1995*). Thus, one might expect a tendency for different introns in the same transcript to use the same mode of initial splice site pairing. Indeed, we observe that there are 5% fewer genes with mixed modes of splicing site recognition than expected by chance (permutation test p<0.001).

Gene architecture may be related to expression dynamics and function. We observed that genes with architectures favoring intron definition were enriched for Gene Ontology categories involved in RNA processing and metabolic processes (*Supplementary file 2*), while genes with predominant exon definition were enriched for developmental processes and stress responses (*Supplementary file 3*). Furthermore, the enrichment within many of the most significant categories of developmental processes and response to oxidative stress increased for genes with increased proportions of exon-defined introns (*Figure 4E*). Timing of gene expression may be of particular

importance during early development, when cell cycles are very short (*Tadros and Lipshitz, 2009*; *Artieri and Fraser, 2014*), and in stress response when protein production must be turned on quickly. Thus, genes in developmental and stress response pathways might have evolved structures that favor exon definition to take advantage of the increased splicing speed and accuracy associated with this mode of recognition, although rates of other steps (e.g. transcription, 3' end formation) will also contribute to expression dynamics and may be rate limiting in some cases. Interestingly, the half-life of the last intron is longer than the expected time to transcribe the last exon in 92% of genes expressed in *Drosophila* S2 cells, making splicing likely to be rate limiting for expression of a large portion of these genes.

We also observed that introns in genes with higher expression levels tended to have faster rates of splicing. This trend could reflect various factors, including differences in the chromatin or nuclear environment of the nascent transcript, or perhaps selection to increase the efficiency of spliceosome recycling. There is evidence that genes with higher expression levels have evolved biased codon usage to enhance translation elongation rates and promote efficient ribosome recycling in order to boost the pool of available ribosomes and optimize overall translation efficiency (*Quax et al., 2015*) (*Kudla et al., 2009*). Similarly, highly expressed genes may have evolved exon definition to boost their speed of splicing and recycling of the splicing machinery in order to boost the pool of free snRNPs and optimize the overall splicing efficiency in the cell. Here, we measured splicing rates in *Drosophila* S2 cells, a highly proliferative cell type that likely has high concentrations of the core gene expression machinery involved in transcription, translation and splicing. Thus, it may be interesting to study splicing rates in other cell types where competition for the splicing machinery might be more pronounced, including cells undergoing developmental transitions or cellular responses that involve rapidly ramping up transcriptional production (*Munding et al., 2013*).

# Materials and methods

## Key resources table

| Reagent type (species) or resource | Designation | Source or reference | Identifiers | Additional information |
|---|---|---|---|---|
| Cell line (*Drosophila melanogaster*) | S2 | Drosophila Genome Resource Center | Stock #6 | |
| Chemical compound | 4-thiouridine (4sU) | Sigma | T4509 | |
| Software | Tophat 2.0.4 | *Trapnell et al. (2009)*; PMID: 19289445 | | |
| Software | Kallisto | *Bray et al., 2016*; PMID: 27043002 | | |
| Software | MaxEntScan | *Yeo and Burge, 2004*; PMID: 15285897 | | |
| Software | MEME | *Bailey et al. (2009)*; PMID: 19458158 | | |
| Software | STAR v2.5 | *Dobin et al. (2013)*; PMID: 23104886 | | |

### Generation of 4sU RNA-seq

*Drosophila melanogaster* S2 cells (Stock Number #6) were obtained directly from the Drosophila Genomics Resource Center (DGRC) in Bloomington Indiana, and maintained at low passage number as indicated by DGRC guidelines. Cells were regularly confirmed to be free of contamination (e.g. mycoplasma) through PCR-based tests as recommended by the NIH.

Newly transcribed RNA from three independent replicates of S2 cells were labeled for 5, 10 and 20 min using 500 µM 4-thiouridine (Sigma, T4509). Additionally, for analysis of steady-state RNA levels, two independent biological replicates of *Drosophila* S2 cells were generated without 4-thiouridine labeling. To normalize samples and assess metabolic labeled RNA capture efficiency, several synthetic RNAs were spiked into the Trizol preparation at specific quantities per $10^6$ cells. Quantities were determined as described previously (*Henriques et al., 2013*). Total RNA was extracted with Trizol (Qiagen, Germany) and treated for 15 min with DNAseI amplification grade (Invitrogen, Carlsbad CA) per manufacturer's instructions. To purify metabolic labeled RNA we used 300 µg total RNA for the biotinylation reaction. Separation of total RNA into newly transcribed and untagged pre-existing RNA was performed as previously described (*Windhager et al., 2012*; *Cleary et al., 2005*). Specifically, 4sU-labeled RNA was biotinylated using EZ-Link Biotin-HPDP (Thermo Fisher, Waltham MA), dissolved in dimethylformamide (DMF) at a concentration of 1 mg/

ml. Biotinylation was done in labeling buffer (10 mM Tris pH 7.4, 1 mM EDTA) and 0.2 mg/ml Biotin-HPDP for 2 hr at 25°C. Unbound Biotin-HPDP was removed by extraction with chloroform/isoamylalcohol (24:1) using MaXtract (high density) tubes (Qiagen, Germany). RNA was precipitated at 20,000 g for 20 min with a 1:10 vol of 5 M NaCl and 2.5X volume of ethanol. The pellet was washed with ice-cold 75% ethanol and precipitated again at 20,000 g for 5 min. The pellet was resuspended in 1 ml RPB buffer (300 mM NaCl, 10 mM Tris pH 7.5, 1 mM EDTA).

Biotinylated RNA was captured using Streptavidin MagneSphere Paramagnetic particles (Promega, Madison WI). Before incubation with biotinylated RNA, streptavidin beads were washed four times with wash buffer (50 mM NaCl, 10 mM Tris pH 7.5, 1 mM EDTA) and blocked with 1% polyvinylpyrrolidone (Millipore Sigma, Burlington MA) for 10 min with rotation. Biotinylated RNA was then incubated with 600 μl of beads with rotation for 30 min at 25°C. Beads were magnetically fixed and washed 5 times with 4TU wash buffer (1 M NaCl, 10 mM Tris pH 7.5, 1 mM EDTA, 0.1% Tween 20). Unlabeled RNA present in the supernatant was discarded. 4sU-RNA was eluted twice with 75 μl of freshly prepared 100 mM dithiothreitol (DTT). RNA was recovered from eluates by ethanol precipitation as described above. As per library preparation, RNA quality was assessed using a Bioanalyzer Nano ChIP (Agilent). Ribosomal RNA was removed prior to library construction by hybridizing to ribo-depletion beads that contain biotinylated capture probes (Ribo-Zero, Epicentre, Madison WI). RNA was then fragmented and libraries were prepared according to the TruSeq Stranded Total RNA Gold Kit (Illumina, San Diego CA) using random hexamer priming. cDNA for the two 'total' RNA samples were prepared using an equal mix of random hexamers and oligo-dT primers.

Libraries were sequenced on an Illumina HiSeq machine with paired-end 51 nt reads (100 nt reads for the 'total' RNA samples), generating an average of 126M read pairs per library. Reads for each sample were filtered, removing pairs where the mean quality score of one or both mates fell below 20. One million pairs were then extracted at random, and aligned to the dm3 reference assembly (4sU RNA-seq, due to enhanced coverage of introns) or an index composed of all FlyBase release 5.57 transcripts (total RNA-seq) using bowtie 0.12.8 (*Langmead et al., 2009*) allowing two mismatches, a maximum fragment length of 10 kb, reporting uniquely mappable pairs only (-m1 –v2 –X10000). Mean fragment length and standard deviation was then assessed using CollectInsertSizeMetrics, a component of Picard Tools 1.62. All reads were subsequently aligned to dm3 with Tophat 2.0.4 (*Trapnell et al., 2009*) utilizing bowtie1 as the underlying aligner, allowing up to 10 reported alignments, passing the fragment length and stdev calculated above, and setting the minimum and maximum intron size to those observed in the FlyBase 5.57 annotations (–bowtie1 –g10 –min-intron-size 25 –max-intron-size 250,000). Strand-specific alignments were performed for the 4sU RNA-seq (–library-type fr-firststrand), while unstranded alignments were performed for the total RNA-seq (–library-type unstranded). Coverage tracks were generated for each using genomeCoverageBed, a component of bedtools v2.16.2 (*Quinlan and Hall, 2010*).

Gene expression values (TPMs) in each replicate library were calculated using Kallisto (*Bray et al., 2016*) and the transcriptome annotations from FlyBase *Drosophila melanogaster* Release 5.57 (*St Pierre et al., 2014*).

## Approaches to estimate the rate of mRNA splicing from 4sU-seq data

We assessed three approaches of estimating mRNA splicing rates, which are intended to be applied to nascent RNA sequencing data after progressive metabolic labeling. All of these approaches rest on two main assumptions: (1) that all transcripts synthesized during the labeling period incorporate the 4sU label; and (2) that all transcripts containing label are equally likely to be captured. The first of these assumptions seem very plausible, while the second is undoubtedly an oversimplification. Additionally, our approaches neglect any contribution from mRNA decay since mRNA half-lives are typically >2 hr in flies, much longer than typical splicing half-lives estimated here (*Herold et al., 2003*). Finally, 4sU uptake and incorporation in mammalian cells has been observed to occur in less than a minute (*Rädle et al., 2013*; *Windhager et al., 2012*), with uptake and incorporation rates depending on the concentration of 4sU in the media and the growth conditions of the cell line. *Drosophila* S2 cells are suspension cells, which have increased 4sU uptake efficiency (*Windhager et al., 2012*), and our data were collected after incubation with a high concentration of 4sU (500 μM). Thus, it is reasonable to expect that 4sU uptake and incorporation in *Drosophila* S2 cells also occurs

rapidly. Any slight lag time could result in a slight overestimation of splicing half-lives using the approaches outlined below.

## Estimating the rate of mRNA splicing using the ratio of intron coverage

Following the approach described in *Windhager et al. (2012)*, we calculated intron ratios between longest labeling periods and shortest labeling periods as a proxy for splicing speed. Specifically:

$$intron\ ratio = \frac{intron\ coverage_{60\ min}}{intron\ coverage_{5\ min}}$$

Since this approach does not specifically estimate intron half-lives, we only performed rank comparisons with our simulated data and estimations from other approaches (as described below). Importantly, this method neglects variability in time since synthesis among introns in different genic positions.

## Estimating the rate of mRNA splicing by modeling the rate of ψ decay

This approach estimates intron half-lives by explicitly modeling data across the labeling periods. To do so, we consider the percent of newly created transcripts that still had intron reads in each labeling period by calculating an intron-specific percent spliced in (PSI or $\Psi$) value, using reads within the intron body and junction reads spanning the 5′ and 3′ splice sites bordering the intron as calculated by the software MISO (*Katz et al., 2010*).

A standard progressive metabolic labeling design will result in the isolation of transcripts that initiated transcription at any time during the labeling period, of duration $L$, as well as transcripts that were elongated during this period but initiated prior to the labeling period. For a transcript to be labeled during the labeling period and include portions informative about the splicing of an intron, the polymerase must have either: (i) transcribed the 3′-most base of the intron during the interval of labeling – between time 0 and $L$ or (ii) have transcribed this base before time 0, but not terminated transcription prior to time 0. The time since synthesis of the intron in case (ii) will be distributed uniformly between $L$ and $L + \tau$, where $\tau$ is the time required to transcribe the portion of the transcript 3′ of the intron. $\tau$ was estimated as the distance $D$ from the 3′-most base in the intron to the annotated transcription endpoint, in nt, divided by an average, constant transcription rate. We therefore estimated the mean time since synthesis for each intron captured, $T_{inferred}$, as

$$T_{\text{inferred}} = \frac{L + \tau}{2}$$

where $L$ is defined by the experimental condition, and $\tau$ is calculated separately for each intron depending on its location in the transcript.

We used the set of $\Psi$ values across inferred lifetimes to estimate the rate at which each intron was excised by fitting a first-order exponential decay model:

$$\Psi(t) = \Psi_0 e^{-\lambda T_{\text{inferred}}}$$

where $t$ is the time since synthesis of the intron, $\Psi(t)$ is the $\Psi$ value at time $t$ (so that $\Psi(0) = \Psi_0 = 1$ by definition), and $\lambda$ is the decay constant. This model was fitted using log-transformed $\Psi$ values for computational efficiency and splicing half-lives were estimated as

$$t_{\frac{1}{2}} = \frac{\log(2)}{\lambda}$$

where $t_{1/2}$ represents the time at which half of the transcripts containing the intron have completed excision of the intron. Note that for mathematical simplicity, this approach models the population of introns captured (which will have varying time since synthesis) by the mean time since synthesis, rather than integrating over the distribution of times as in the junction dynamics approach below. Although this approach yields more quantitative half-life measurements than the intron ratio approach, the inclusion of reads deriving from incompletely synthesized introns may inflate $\Psi$ values and therefore lead to over-estimation of intron half-lives, especially for long introns at short labeling periods, and use of a mean time since synthesis is an approximation.

# Estimating the rate of mRNA splicing by mathematical modeling of junction ratios

One limitation of our modeling of PSI decay is the inability to distinguish reads deriving from incompletely transcribed introns from those deriving from completed but unspliced introns, both of which are captured by sequencing of 4sU-labeled nascent RNA. Our progressive labeling experimental design yields three populations of transcripts: partial-intron transcripts which do not contain a completely transcribed intron, unspliced transcripts that have complete introns that have not been spliced, and spliced transcripts. This spectrum can be conceptualized as a range of endpoints of a polymerase that could potentially contribute reads to the intron (*Figure 1A*) – specifically, if a polymerase is in the body of the intron at the end of the labeling period, it would produce a labeled transcript that includes only a portion of the intron. If the polymerase is further along the gene body but has not completed transcription of the full gene, it would produce a labeled transcript that will either include the intron or not, depending on splicing status. If the transcript is completely transcribed within the labeling period, it would also have a probability of either including the intron or not depending on splicing status. While, in reality, the associated polymerase for these completed transcripts would disengage from the DNA shortly following cleavage of the mRNA, conceptually it can be associated with the point where it would be at the end of the labeling period had it continued transcribing without dropoff past the end of the gene, so that all considered polymerases can be associated to positions on a line.

Considering the end of the intron to be coordinate 0, the range of endpoints relevant to the intron is [-*i*, *e* + *d* + *Lr*], where *i* is the length of the intron, *e* is the length of the downstream exon, *d* is the distance from the end of the downstream exon to the polyA site at the end of the gene, *L* is the labeling period, and *r* is the rate of transcription (again assumed to be constant across the genome). Before position -*i*, the resulting transcript will have no overlap with the intron, and after position *e* + *d* + *Lr*, the resulting transcript cannot have incorporated the metabolic label. Transcripts associated with these polymerase end points can be divided into the three populations they come from: pre-spliced transcripts, unspliced transcripts, and spliced transcripts (as described above). Short-read sequencing makes it difficult to determine which reads come from which transcript population. The only reads that can be definitively assigned are junction reads, where intron-exon junction reads come from the unspliced category and exon-exon junction reads come from the spliced category.

We can consider the expected contribution to the unspliced category by a polymerase at the transcript level by letting the probability of any endpoint *x* be some value *p*, a constant across the region [-*i*, *e* + *d* + *Lr*]. Assuming the region is continuous, the contribution to the region can be considered as a Boolean variable: one if the transcript is spliced, and 0 otherwise. We can model the probability of splicing as an exponential decay process with some half-life *h* (where *h* is equivalent to $t_{1/2}$), written as $2^{-\frac{x}{hr}}$, since the probability of being unspliced should be 1 when $x = 0$ and $\frac{1}{2}$ when *x* is distance *hr* past the end of the intron (where $x = hr$). Finally, we can integrate over the polymerase positions in a gene locus to get a proportional contribution of each to unspliced reads. More formally:

$$
\begin{aligned}
E[\textit{unspliced contribution}] &= \int_0^{e+d+Lr} p\left(2^{-\frac{x}{hr}} \cdot 1 + \left(1 - 2^{-\frac{x}{hr}}\right) \cdot 0\right) dx \\
&= p\left[\int_0^{e+d+Lr} 2^{-\frac{x}{hr}} dx\right] \\
&= p\left[\frac{hr}{\log(2)}\left(1 - 2^{-\frac{e+d+Lr}{hr}}\right)\right]
\end{aligned}
$$

Similarly, we can consider the expected contribution to the spliced category by a polymerase within this framework by flipping the Boolean assignment in the region [*0*, *e* + *d* + *Lr*]:

$$
\begin{aligned}
E[\textit{spliced contribution}] &= \int_0^{e+d+Lr} p\left(2^{-\frac{x}{hr}} \cdot 0 + \left(1 - 2^{-\frac{x}{hr}}\right) \cdot 1\right) dx \\
&= p\left[\int_0^{e+d+Lr}\left(1 - 2^{-\frac{x}{hr}}\right) dx\right] \\
&= p\left[e + d + Lr - \frac{hr}{\log(2)}\left(1 - 2^{-\frac{e+d+Lr}{hr}}\right)\right]
\end{aligned}
$$

To obtain expressions for the expected number of both intron-exon and exon-exon junction reads, we make two assumptions: (1) each transcript is spliced independently and (2) there is an

equal and constant probability of sampling a junction read from a spliced transcript as from an unspliced transcript. Under these assumptions, the expressions derived above can be multiplied by the total number of polymerases (which depends on the expression level) and the probability of observing a junction. Taking the ratio of these new expressions and assuming a constant transcription rate yields a first-order approximation of the expected ratio of the two different kinds of junction reads as a function of the half-life:

$$\frac{no.\ of\ intron-exon\ junction\ reads}{no.\ of\ exon-exon\ junction\ reads} \approx \frac{\frac{hr}{\log(2)}\left(1-2^{-\frac{e+d+Lr}{hr}}\right)}{e+d+Lr-\frac{hr}{\log(2)}\left(1-2^{-\frac{e+d+Lr}{hr}}\right)}$$

To simplify the expression, we introduce the substitutions $L^{'} = e + d + Lr$ and $h^{'} = \frac{hr}{\log(2)}$:

$$= \frac{h^{'}\left(1-e^{-\frac{L^{'}}{h^{'}}}\right)}{L^{'}-h^{'}\left(1-e^{-\frac{L^{'}}{h^{'}}}\right)}$$

$$= \frac{1}{\frac{L^{'}}{h^{'}}\left(1-e^{-\frac{L^{'}}{h^{'}}}\right)-1}$$

Solving this expression for $h$ yields an estimate for the half-life of an intron. Since this equation has no analytical solution, we solved this expression numerically using the general-purpose optimization algorithm (*optim* in the R statistical environment) to minimize error in the value of $h$. Data from different labeling periods were fit jointly to incorporate variation in junction read capture across labeling periods.

## Simulations to assess modeling approaches

We used simulations to assess the accuracy of each model. To do so, we simulated data for an intron across a range of different genomic and expression contexts: intron length $i$ (40nt – 100 kb), downstream distance $d$ (1–5 kb), expression level in TPM (1–46 TPM), labeling periods $L$ (5, 10, 20, and 60 min) and half-lives $h$ (0.2–100 min). These parameters recapitulate standard distributions of gene structures, expression values, and experimental conditions. To generate read data from transcripts, we simulated several steps in nascent RNA transcription, capture, and library preparation.

In order to generate transcripts, we assumed a constant total number of transcripts (100 million) and sampled transcripts at specified TPM values. For instance, for a TPM = 5, 500 transcripts would be sampled for a total pool of 100 million. The upstream distance was held constant at 500 nt. Transcripts were generated by selecting their endpoints uniformly from the region [-$i$, $e + d + Lr$], where $i$, $d$, and $L$ were varied (in the ranges specified above), $r$ was a constant 1500 nt/min, and $e$ was fixed at 300 nt. For transcripts with end points beyond 0, the splicing state was probabilistically determined based on the simulated half-life and end point. Transcripts with end points beyond $e + d$ were terminated at $e + d$, but the longer end points affected the probability of splicing since they represent transcripts that were completed before the end of the labeling period.

The set of final transcripts with variable lengths (depending on simulated parameters, the distribution of end points, and splicing state) were fragmented with fragment lengths drawn from a modified Weibull distribution with $\delta = \log_{10}(\text{length})$ and $\eta = 200$ nt as described in *Griebel et al. (2012)*. This procedure consists of sampling $n - 1$ breakpoints uniformly from the interval [0,1] for each transcript, where

$$n = \frac{\text{length}}{\eta\Gamma\left(1+\frac{1}{\delta}\right)}$$

These breakpoints then give $n$ length fractions adding up to 1: $x_1$, …, $x_n$, which can be translated into fragments of the transcript with lengths $d_i$ using the transformation:

$$d_i = \text{length}\frac{x_i^{\frac{1}{\delta}}}{\left(\sum_j x_j^{\frac{1}{\delta}}\right)}$$

Finally, the resulting fragments are size-selected, keeping only those in the range [200, 300] nt, and recording the first 50 nt of these fragments as our simulated reads.

Correlations between estimated and simulated half-lives were calculated across half-lives for each parameter combination. Absolute error was calculated as the absolute value of the ratio between the estimated half-life for intron $i$ and the simulated half-life for intron $i$:

$$\left| \log_2 \frac{estimated\ t_{\frac{1}{2}}\ for\ intron\ i}{simulated\ t_{\frac{1}{2}}\ for\ intron\ i} \right|$$

In order to compare magnitudes assigned by all three methods (since the intron ratio measure does not assign an explicit half-life), we compared fold differences in half-lives between pairs of introns (intron $j$ and intron $k$) estimated by each of the three methods to the fold differences used in the simulations:

$$\left| \log_2 \frac{\frac{estimated\ t_{\frac{1}{2}}\ for\ intron\ j}{estimated\ t_{\frac{1}{2}}\ for\ intron\ k}}{\frac{simulated\ t_{\frac{1}{2}}\ for\ intron\ j}{simulated\ t_{\frac{1}{2}}\ for\ intron\ k}} \right|$$

## Estimating splicing rates from 4sU-RNA-seq data

To measure the efficiency at which each intron was spliced out in *Drosophila* S2 cells, we considered a subset of introns for which we had sufficient power to measure splicing rates using our junction ratio modeling. For all following analyses, we only considered the 25,576 introns that met the following conditions: (1) in a gene with TPM >5 in the total RNA libraries, (2) was completely or predominantly spliced out in the total RNA libraries ($\Psi$ <0.2, as estimated by MISO [*Katz et al., 2010*]), (3) had at least 1 intron-exon junction read in the 5-min labeling period, since introns without any intron-exon junction reads at the earliest labeling period are likely spliced too fast to be detected in our data, (4) had at least 1 exon-exon junction read in all of the labeling periods (to avoid non-zero ratios), and (5) did not contain an annotated alternative splicing event such as a skipped exon. Introns flanking annotated cassette exons (with $\Psi$ >0.95 in S2 cells) were retained for further analysis.

The majority of mRNA introns are spliced out using the major spliceosome (including the U2 snRNP), but a small proportion of introns are spliced out using the minor splicesome, which uses recognition of the 3′ splice site by the U12 snRNP. Previous studies have found that U12-type introns might be spliced slower than U2-type introns, suggesting that U12-dependent mechanisms have different kinetic considerations. In our dataset, there were just 7 U12-type introns in genes expressed in S2 cells with sufficient read depth for us to calculate splicing half-lives: '*chrX:9231925–9232086:-*', '*chr3R:22789242–22789840:-*', '*chr2R:14708343–14708546:+*', '*chr3R:20895772–20895925:+*', '*chr2L:16743570–16743760:+*', '*chr3R:4838027–4838559:-*', '*chrX:17518238–17519563:-*'. Consistent with previous observations, these seven introns are, on average, spliced more slowly than U2-type introns (median half-life for U12 introns = 4.9 min; median half-life for U2-type introns = 1.9 min), but this difference is not statistically significant because of the small numbers involved. These U12-type introns were somewhat longer than U2-type introns expressed in S2 cells (median length for U12-type introns = 203 nt; median length for U2-type introns = 69 nt, not significant). Given the low numbers and lack of statistical power or significance, we do not discuss these U12-type introns in further detail.

For each of the introns retained for further analyses, junction reads (intron-exon and exon-exon) were extracted using pysam, conditional on at least (1) a 10nt overlap with both the intron and downstream exon regions for intron-exon junction reads and (2) a 10nt overlap with both upstream and downstream exon regions for exon-exon junction reads. Junction reads were combined across the three replicates from each labeling period to increase power to model half-lives. Splicing half-lives were estimate using the junction ratio method (as described above) with a constant *Drosophila* transcription rate of 1500 nt/min, derived from the literature (*Ardehali and Lis, 2009*; *Garcia et al., 2013*). Confidence intervals for each intron half-life were estimated by subsampling the total read pool for each replicate to 70% of the data and bootstrapping intron half-lives across 10 samples. Goodness-of-fit for the junction dynamics model was assessed using a residual sum of squares

comparing the empirical junction ratio $R_{emp}$ with the estimated junction ratio $R_{est}$ across labeling periods $L$ where $L$ = 5, 10, and 20 min:

$$residual\ sum\ of\ squares = \sum_L \left(R_{emp} - R_{est}\right)^2$$

## Calculating splice site scores

We calculated the strength of splice sites using a maximum entropy model as implemented in maxEntScan (*Yeo and Burge, 2004*) using 9 bp around the 5' splice site (−3: +6) and 23 bp around the 3' splice site (−20:+3). These models were optimized on mammalian splice site preferences, but seem to be reasonable also for *Drosophila* and have been used in gene prediction in fly genomes.

## Identifying branch point positions

To identify branch point motifs in short introns, we first derived a branch site position weight matrix by running MEME (*Bailey et al., 2009*) on regions between 15 and 45 nt upstream of the 3' splice site for 10,000 randomly sampled *Drosophila melanogaster* introns (as described in [*Lim and Burge, 2001*]). The branchpoint for each intron was identified as the −3 'A' position of the highest scoring motif in the 50 nt region upstream of the 3' splice site. Sequence logos for each length bin were created using the PICTOGRAM program (http://genes.mit.edu/pictogram.html).

## Calculating *kmer* enrichment

We used a custom python script to identify 6mers enriched in regions in and around intron-defined or exon-defined introns. Upstream exons were defined as the flanking exon upstream of the given intron, downstream exons were defined as the flanking exon downstream of the given intron, and intronic regions were defined as the space between +10 from the 5' splice site and −30 of the 3' splice site for the given intron. Briefly, the number of occurrences of each 6mer in the region were counted (normalized for variable lengths of regions), and significantly enriched 6mers were identified using the Fisher's exact test. The resulting p-values were then corrected using the Benjamini-Hochberg procedure.

## Estimating splicing accuracy

The accuracy of splicing in *Drosophila* introns was estimated by identifying non-annotated junction reads with non-canonical splice site sequences within annotated introns. To do so, we first remapped the raw 4sU-seq reads with the STAR v2.5 short-read mapper (*Dobin et al., 2013*), with the mapping parameter *–outSAMattribute NH HI AS nM jM* to mark the intron motif category of each junction read in the final mapped file.

The *jM* attribute adds a *jM:B:c* SAM attribute to all reads arising from exon-exon junctions. All junction reads were first isolated and separated based on the value assigned to the *jM:B:c* tag. Junction reads with splice sites in the following categories were considered to be annotatedor canonical: (1) any annotated splice site based on the FlyBase *D. melanogaster* Release 5.57 gene structures [*jM:B:c, 20–26*], (2) intron terminal nucleotides containing 'GT-AG' (or the reverse complement) [*jM:B:c, 1* or *jM:B:c, 2*], (3) intron terminal dinucleotides containing 'GC-AG' (or the reverse complement) [*jM:B:c, 3* or *jM:B:c, 4*], and (4) intron terminal dinucleotides containing 'AT-AC' (or the reverse complement) [*jM:B:c, 5* or *jM:B:c, 6*]. Junction reads with *jM:B:c, 0* were considered to arise from non-canonical non-annotated splice sites. We calculated the frequency of inaccurate splice junctions for each intron as a ratio of the density of reads arising from non-canonical non-annotated splice sites to the density of all junction reads from the intron.

## Identifying enhancer regions

We used *Drosophila* transcriptional enhancers defined by the STARR-seq enhancer testing assay described in *Arnold et al. (2013)*. Significant STARR-seq peaks were overlapped with annotated *Drosophila* introns to identify transcriptional enhancers located within introns.

## Modeling major contributors to splicing rate variability

To estimate the extent to which factors other than intron length accounted for variance in splicing rates, we fit a linear model of the following form to all non-first introns $i$, separately for intron-defined and exon-defined introns:

$$y_i = \beta_0 + \beta_0 + \beta_{i1}x_{i1} + \beta_{i2}x_{i2} + \beta_{i3}x_{i3} + \beta_{i4}x_{i4} + \beta_{i5}x_{i5} + \beta_{i6}x_{i6} + \beta_{i7}x_{i7} + \beta_{i8}x_{i8} + \beta_{i9}x_{i9}$$
$$+ \beta_{i10}x_{i10} + \beta_{i11}x_{i11} + \beta_{i12}x_{i12} + \beta_{i13}x_{i13} + \beta_{i14}x_{i14} + \varepsilon_i$$

where $y_i\log10$ are half-lives ($log_{10}$, minutes), $x_{i1}\log10$ is intron length ($log_{10}$, nt), $x_{i2}\log10$ is position of intron in a transcript, $x_{i3}\log10$ is gene expression ($log_{10}$, TPM), $x_{i4}\log10$ is an indicator for the presence of an enhancer in intron $i$ (based on STARR-seq), $x_{i5}\log10$ is 5' splice site score (from maxEnt), $x_{i6}\log10$ is 3' splice site score (from maxEnt), $x_{i7}\log10$ is the length of the first intron of the transcript ($log_{10}$, nt), $x_{i8}\log10$ is the half-life of the first intron of the transcript ($log_{10}$, min), $x_{i9}\log10$ is an indicate for the presence of an enhancer in the first intron of the transcript (based on STARR-seq), $x_{i10}\log10$ is the length of the upstream exon ($log_{10}$, nt), $x_{i11}\log10$ is the length of the downstream exon ($log_{10}$, nt), $x_{i12}\log10$ the % of A + U nt in intron $i$ (excluding the splice site region), $x_{i3}\log10$ is the % of A + U nt in the 3' region of intron $i$ ($-40{:}-21$ from 3' splice site), $x_{i4}\log10$ is the % of A + U nt in the 5' region of intron $i$ (+7 from 5' splice site to $-41$ from the 3' splice site).

We used the values from this multiple linear regression model to estimate the relative importance of each parameter contributing to variance in splicing rates. To do so, we used the *relaimpo* package in the R statistical environment (**Grömping, 2006**), which arrives at a relative importance percentage by averaging the sequential sum-of-squares obtained from all possible orderings of the predictors in the model.

## Gene-level analyses of splicing rates

Introns within a gene were defined as all introns annotated as being from the same parent gene (regardless of annotated transcript classifications). Genes were classified into splice site recognition mode categories if at least two thirds of their introns had consistent RIME values, where RIME <1 was classified as intron-definition and RIME >= 1 was classified as exon-definition for this analysis. The remaining genes were classified as mixed-definition. To calculate an expected number of intron- or exon-defined genes, we performed a weighted sampling introns from each gene (matching the distribution of number of introns per gene) and classifying them as either intron-defined or exon-defined, with the weights determined by the overall proportion of introns with RIME less than or greater than one, respectively. Each sampled gene was then defined as having intron- or exon-, or mixed-definition as described above. This procedure was done 1000 times to arrive at a distribution of expected classifications.

## Gene ontology analyses

All Gene Ontology analyses were performed using the clusterProfiler package (**Yu et al., 2012**) in the R statistical software environment and biological processes gene ontology categories. Gene ontology analyses for fast and slow genes were also performed. Regardless of the predominant mode of splice site definition, we observed that genes had different intrinsic average splicing rates. To discriminate between genes that have overall fast or overall slow splicing, we used the average of all intron half-lives within a gene and selected the genes in the 10[th] percentile for short and long average half-lives. Genes with the fastest splicing were enriched for cytoplasmic translation and biosynthetic processes and are highly expressed (median TPM = 84). Genes with the slowest splicing were enriched for metabolic processes, catabolic processes, and RNA processing functions and are less highly expressed (median TPM = 31). Since these enrichments and expression patterns are similar to the distinction between mostly intron- or mostly exon-defined genes, we looked to see if the fastest genes are more likely to be mostly exon-defined. Indeed, 8% of the fastest spliced genes are mostly exon-defined (compared to 3% of the slowest genes) and an average of 28% of the introns in the fastest genes are likely exon-defined (compared to an average of 9% of introns in the slowest genes).

## Data availability

Sequencing data have been deposited at the Gene Expression Omnibus (GEO) database under accession GSE93763.

## Code availability

Source code for simulations, the junction dynamics approach, analyses of splicing rates and splice site recognition mode, and to recreate figures from this manuscript is available at https://github.com/athma/splicingrates (*Pai, 2018*; copy archived at https://github.com/elifesciences-publications/splicingrates).

## Acknowledgements

We thank members of the Burge lab, Brent Graveley, Scott Roy and Phil Sharp for helpful comments on earlier versions of this manuscript. This work was supported by a Jane Coffin Childs Postdoctoral Fellowship (AAP), by DOE Computational Science Graduate Fellowship DE-FG02-97ER25308 (KM), by NIH grant R01-GM085319 (CBB), and by the Intramural Research Program of the National Institutes of Health, National Institute of Environmental Health Sciences to KA (Z01 ES101987).

## Additional information

### Competing interests

Karen Adelman: Reviewing editor, *eLife*. The other authors declare that no competing interests exist.

### Funding

| Funder | Grant reference number | Author |
| --- | --- | --- |
| National Institutes of Health | Z01-ES101987 | Telmo Henriques<br>Adam Burkholder<br>Karen Adelman |
| National Institutes of Health | R01-GM085319 | Athma A Pai<br>Christopher B Burge |
| Jane Coffin Childs Memorial Fund for Medical Research | | Athma A Pai |
| U.S. Department of Energy | FG02-97ER25308 | Kayla McCue |

The funders had no role in study design, data collection and interpretation, or the decision to submit the work for publication.

### Author contributions

Athma A Pai, Software, Formal analysis, Investigation, Visualization, Methodology, Writing—original draft; Telmo Henriques, Investigation, Writing—review and editing; Kayla McCue, Software, Formal analysis, Visualization, Methodology, Writing—review and editing; Adam Burkholder, Data curation, Software; Karen Adelman, Conceptualization, Supervision, Funding acquisition, Writing—review and editing; Christopher B Burge, Conceptualization, Supervision, Funding acquisition, Writing—original draft

### Author ORCIDs

Athma A Pai [iD] http://orcid.org/0000-0002-7995-9948
Karen Adelman [iD] http://orcid.org/0000-0001-5364-334X
Christopher B Burge [iD] http://orcid.org/0000-0001-9047-5648

### Decision letter and Author response

Decision letter https://doi.org/10.7554/eLife.32537.021
Author response https://doi.org/10.7554/eLife.32537.022

## Additional files

### Supplementary files

• Supplementary file 1. Summary of introns analyzed. *Column 1 – intron:* Coordinates of introns, with chr:start:end:strand for the upstream flanking exon and the chr:start:end:strand for the downstream flanking exon separated with a '@'. *Column 2 – gene:* FlyBase gene symbol for parent gene. *Column 3 – TPM:* Gene expression values calculated with kallisto. *Column 4 – PSI:* MISO-derived Ψ values of intron in total time point (average across two replicates). *Column 5 – intron_position:* Position of intron relative to other introns in the transcript. *Column 6 – intron_length:* Length of intron (nucleotides). *Column 7 – intron_type:* Regulatory type of intron, where CI is constitutively spliced intron, RI is an annotated retained intron, and SEflanking is an intron that flanks a retained intron. *Column 8 – ss5_maxEnt:* maxEnt-derived splice site score for the 5' splice site of the intron. *Column 9 – ss3_maxEnt:* maxEnt-derived splice site score for the 3' splice site of the intron. *Column 10 – contains_enhancer:* Flag for whether the intron contains a transcriptional enhancer as defined by STARR-seq. *Column 11 – upexon_length:* Length of upstream exon (nucleotides). *Column 12 – downexon_length:* Length of downstream exon (nucleotides). *Column 13 – three_length:* Length of the region from the 3' splice site of the intron to the polyA site of the transcript. *Columns 14–16 – ie_count_[timepoint]:* count of intron-exon junction reads for each of the labeling periods (summed across three replicates per labeling period). *Columns 17–19 – ee_count_[timepoint]:* count of exon-exon junction reads for each of the labeling periods (summed across three replicates per labeling period). *Column 20 – halflife:* Half-life of intron computed using the junction dynamic approach. *Column 21 – halflife_error:* Standard error around the half-life estimate, derived from bootstrapping the half-life across subsampled populations of reads. *Column 22 – accuracy:* percent of junction reads from unannotated, non-canonical splice sites within the intron
DOI: https://doi.org/10.7554/eLife.32537.014

• Supplementary file 2. Gene Ontology for mostly intron-defined genes. Summary output from clusterProfiler for significantly enriched biological process gene ontology categories.
DOI: https://doi.org/10.7554/eLife.32537.015

• Supplementary file 3. Gene Ontology for mostly exon-defined genes. Summary output from clusterProfiler for significantly enriched biological process gene ontology categories.
DOI: https://doi.org/10.7554/eLife.32537.016

• Transparent reporting form
DOI: https://doi.org/10.7554/eLife.32537.017

### Major datasets

The following dataset was generated:

| Author(s) | Year | Dataset title | Dataset URL | Database, license, and accessibility information |
|---|---|---|---|---|
| Pai AA, Henriques T, McCue K, Burkholder A, Adelman K, Burge CB | 2017 | Drosophila S2 cell 4sU RNA-seq data | https://www.ncbi.nlm.nih.gov/geo/query/acc.cgi?acc=GSE93763 | Publicly available at the NCBI Gene Expression Omnibus (accession no: GSE93763) |

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
