## [Decision Letter]

Thank you for submitting your article "The kinetics of pre-mRNA splicing in the *Drosophila* genome: influence of gene architecture" for consideration by *eLife*. Your article has been reviewed by three peer reviewers, and the evaluation has been overseen by Timothy Nilsen as the Reviewing Editor and James Manley as the Senior Editor. The following individuals involved in review of your submission have agreed to reveal their identity: Andrew G Clark (Reviewer #2); Manuel Irimia (Reviewer #3).

The reviewers have discussed the reviews with one another and the Reviewing Editor has drafted this decision to help you prepare a revised submission.

All of the reviewers were quite positive about the work; all agreed that it provided substantive new insight into splicing kinetics in vivo. All also agreed that the paper was in principle suitable for publication in *eLife*. Nevertheless, each referee raised a few, largely overlapping, concerns that must be addressed via revision. In this regard, it was agreed that these points could be dealt with by revisions to the text both acknowledging some potential caveats to interpretation of the data and explanation of some items that were not clear. In lieu of providing a list of essential revision, we are providing you with the complete reviews with the understanding that all points raised will be thoroughly addressed.

*Reviewer #1:*

Overall, this is an excellent paper. The careful use and description of the simulations is to be commended. The authors make several interesting observations about rates and accuracy of exon defined and intron defined splicing events. Their results are certainly the most complete treatment of gene structure and splicing in *Drosophila*. There are, however, a few issues that should be addressed.

1) The analysis makes the explicit assumption that transcription rates are uniform across genes. This is probably close to true for *Drosophila* but I would think that this could be tested using the data from these experiments and earlier ones from, for example, the Lis lab. I would be most concerned about how non-homogeneous transcription rates might affect some of the conclusions made regarding effects at the ends of genes.

2) A related issue is the unstated but important assumption that 4s-U labeling is instantaneous and uniform. There must be a lag, and perhaps a significant lag, in labeling, particularly in the 5-minute time point, during which the uridine pool is reaching a new equilibrium. There is significant literature on this issue in cultured cells (although perhaps not in S2 cells) from which to make reasonable approximations. From what I can tell, this effect was not modeled in the simulation and it is not clear to what extent this might affect the rates of splicing observed. While most of the analyses in the paper are comparative rather than absolute, the authors do make claims about real-time rates. They should address this point and perhaps show that their measured rates are valid when this issue is included.

3) The discussion of developmental and stress response genes being enriched for exon definition does not seem complete. Exon definition genes are longer and so take longer to transcribe. Splicing appears to be faster for such genes but what is the actual rate limiting step in their synthesis? It could even be 3' end formation rather than elongation or splicing.

4) The authors note that the first introns are often more slowly spliced than subsequent introns. What about last introns and, in particular, exon defined last introns that may rely on different mechanisms?

*Reviewer #2:*

This paper was a delight to review! By labeling nascent RNA transcripts with 4-tiouridine, the authors were able to infer which transcripts retained introns and which had already spliced out the introns. They allowed for modeling of the full dynamics of the splicing process to estimate rates of splicing of every intron in the transcriptome. The goal was to learn how intron and exon lengths impact splicing dynamics. The find that exon-spanning introns are more efficiently spliced than intron-spanning splice sites, and suggest that genes that require rapid activation (such as stress response) have evolved the exon-spanning definition. Also genes that are very highly expressed tend to use the exon definition mode, consistent with the need for faster transcript processing.

The paper is well written and very well motivated. Intron-defined and exon-defined splice sites are clearly spelled out, and the unknowns regarding relative use of these splicing kinetic pathways drive the need for studies like this. The use of 4-thiouracil (4sU) incorporation provided a means to estimate the half-life for the splicing time of essentially every intron in the 5600 most highly expressed genes.

The authors do a good job showing how the method will work in silico, through simulations of the 4sU labeling processes and the inferred splicing half-lives of introns obtained from labeled transcripts with and without introns spliced out.

By doing 5, 10, and 20 min labeling with 4sU, followed by RNA sequencing and supplementing this with RNA-seq data of mature transcripts. The relative ratio of intron-exon junction reads to exon-exon junction reads was highest at the 5 min labeling period and decreased rapidly to low levels at longer times The method would appear at first to rely heavily on the assumption that all transcription proceeds at 1.5 kb/min, a figure that has good empirical support for the average, but which likely varies considerably. Simulations do an adequate job showing that the splicing half-lives are nevertheless well estimated over a reasonably broad range of transcription rates (although presumably if transcription rate were related to splicing progress in some aberrant way, the method could falter).

The data presented here clearly supports the notion that short introns that have relatively long flanking introns will tend to employ the intron definition splicing mode, whereas those with longer introns than the average of the flanking exons will tend to employ the exon definition splicing mode.

One technical detail could be made a bit more clear. The Materials and methods section succeeds very well in describing the way that splicing kinetics are learned from the relative abundances of spliced and unspliced introns in transcripts of different lengths. The contrast of several methods for quantitative analysis is good, and makes the logic clear. But the way that the data provide inference of the distinction between the exon-defined and intron-defined splicing could be more clearly spelled out. It appears to be based solely on RIME value, and while this is a sensible starting point, one would like to know if there is any other confirmation of these calls. In other words, one would like to know error rates for this inference.

An overall assessment of goodness-of-fit of the decay model to data would be good to see.

Given the unusually slow kinetics of splicing of the first intron (which often hold enhancer elements), it does appear to be sensible to drop the first intron in the analysis of the factors that determine splice mode.

Arguments that the splicing modes are a product of natural selection are speculative and are not based on either comparative (interspecific) analysis or on population genetic analysis. The former seems more likely to be feasible, but would entail a radical increase in the work (and so is not recommended here!). And the latter cannot be done without a battery of S2-like cells from different lines, which don't exist. Probably the relatively low frequency of polymorphism in splice sites would erode the power of any population genetic approach. So the suggestion is to make clear in the wording about the claims of selection that these are speculative inferences not based on specific evidence of past action of selection. To this reviewer, this lack does not erode the value of the paper, and the speculative remarks about natural selection do help the reader understand the nature of the question and the meaning of the results.

The splicing dichotomy poses an evolutionary puzzle that might warrant some discussion. Namely, the data appear to support the idea that the splicing modes driven by the relative lengths of introns and flanking exons. But natural selection in turn can also adjust these parameters. So, is there any adaptive argument to be made why *Drosophila* are balanced between the use of both modes?

It is somewhat of a limitation that the entire study was done in S2 cells. Is there any hope of getting 4sU into whole organisms to get at tissue-specific characteristics?

There are a few issues that are not addressed and might be of interest. *Drosophila* make great use of alternatively spliced exons and yet alternative splicing gets very short mention in the paper. Similarly nested genes would be expected to have odd splicing dynamics, and it might be a good sanity check to see this. I expected to see more discussion of the super short introns (<50 bp) found in *Drosophila* – are they all intron-defined and super-rapidly spliced? Is it necessary to invoke different splicing machinery for these?

*Reviewer #3:*

Pai and co-workers investigated genome-wide splicing rates in *Drosophila* S2 cells and how these rates relate to different modes of splicing (i.e. intron and exon definition). For this, the authors employed metabolic labeling coupled to RNA sequencing, combined with elegant mathematical modeling. The study reports several interesting results, including a local maximum for splicing rates for 60-70 nt long introns (the most common length in *Drosophila*) and the unexpected finding that exons surrounded by very long introns are spliced the fastest and most accurately.

Overall, I enjoyed the manuscript and I do not have any major concern. The following are specific comments/suggestions:

1) The fact that introns with similar rates tend to more often co-occur within the same genes seems to make sense. However, I wonder whether this could be driven, at least in part, by the different elongation rates among genes. If I understood it correctly, the model they developed assumes a single elongation rate for all genes. Therefore, it is possible that some gene-level biases are introduced because of this. While this should not affect most of the conclusions in the study, the bias may be strong enough as to create patterns of significant co-occurrence within genes (all introns within a gene will have the same bias and this will be different to most other introns). This is of course difficult to test. Perhaps the authors could use their data, consisting of three time points, to roughly estimate elongation rates and group genes based on these. Irrespectively, I may be a good idea to add a brief note about it.

2) Figure 4: given the results in Figure 4, it would be good to also have a comparison of half-life SDs for randomly sampled introns within groups of similar lengths, to see if this is the main feature driving the signal (rather than gene co-occurrence).

3) Subsection “Exon definition is associated with faster and more accurate splicing” and Figure 3: this plot and the associated description in the text are not very easy to understand. The reference to stripes is ambiguous. I do not have a better suggestion on how to represent the data, but at least the wording in the text could be improved. It may help to label/highlight the sections of interest in the plot.

---

## [Author Response]

Reviewer #1:Overall, this is an excellent paper. The careful use and description of the simulations is to be commended. The authors make several interesting observations about rates and accuracy of exon defined and intron defined splicing events. Their results are certainly the most complete treatment of gene structure and splicing in Drosophila. There are, however, a few issues that should be addressed.1) The analysis makes the explicit assumption that transcription rates are uniform across genes. This is probably close to true for Drosophila but I would think that this could be tested using the data from these experiments and earlier ones from, for example, the Lis lab. I would be most concerned about how non-homogeneous transcription rates might affect some of the conclusions made regarding effects at the ends of genes.

The assumption made regarding transcription rates is supported by earlier studies specifically measuring transcription elongation rates (Adelman et al., 2002; Ardehali and Lis, 2009). Elongation rates are very difficult to assess with our own data. The progressive labeling strategy used here does not incorporate a treatment to synchronize RNA polymerase initiation nor does it directly measure the distribution of RNA polymerases across a gene (our approach integrates across all possible locations), making it hard to calculate region-specific elongation rates across a gene. To the extent that different elongation rates occur within or between genes, we do not expect this to have a very large impact on magnitudes of estimate splicing rates, as long as elongation rates fall within the range of 0.5 kb/min to 4 kb/min (as shown in Figure 1—figure supplement 3). We now explicitly acknowledge that different transcriptional elongation rates, or non-uniform rates across a gene, might have an effect on absolute and/or relative measurements of splicing rates in subsection “Measuring rates of splicing for 25,000 *Drosophila* introns”.

2) A related issue is the unstated but important assumption that 4s-U labeling is instantaneous and uniform. There must be a lag, and perhaps a significant lag, in labeling, particularly in the 5-minute time point, during which the uridine pool is reaching a new equilibrium. There is significant literature on this issue in cultured cells (although perhaps not in S2 cells) from which to make reasonable approximations. From what I can tell, this effect was not modeled in the simulation and it is not clear to what extent this might affect the rates of splicing observed. While most of the analyses in the paper are comparative rather than absolute, the authors do make claims about real-time rates. They should address this point and perhaps show that their measured rates are valid when this issue is included.

There is a potential lag time in 4sU uptake, phosphorylation, and incorporation following the addition of 4sU to the culture media. However, as the reviewer notes, most previous literature is in yeast (where 4sU uptake is slow, (Miller et al., 2011)) and mammalian cells (where 4sU uptake is faster). Mammalian cells likely incorporate 4sU in less than a minute (Rädle et al., 2013; Windhager et al., 2012), though the uptake and incorporation rates are thought to be dependent on the concentration of 4sU in the media and the growth conditions of the cell line. *Drosophila* S2 cells are suspension cells, which have increased 4sU uptake efficiency (Windhager et al., 2012), and our data was collected after incubation with a high concentration of 4sU (500 μM). Thus, it is reasonable to expect that 4sU uptake and incorporation in *Drosophila* S2 cells occurs in less than a minute. Therefore, our assumption that 4sU incorporation is effectively instantaneous may lead to a moderate overestimation of absolute splicing half-lives but is unlikely to have large effects on absolute estimated rates or to impact relative splicing rates between introns. These considerations are now discussed in subsection “Approaches to estimate the rate of mRNA splicing from 4sU-seq data”.

3) The discussion of developmental and stress response genes being enriched for exon definition does not seem complete. Exon definition genes are longer and so take longer to transcribe. Splicing appears to be faster for such genes but what is the actual rate limiting step in their synthesis? It could even be 3' end formation rather than elongation or splicing.

Our analysis does not specifically identify the rate-limiting step for formation of mature mRNA, which is likely to be transcription for very long genes, and RNA processing for short genes. However, we note that in 92% of fly genes, the estimated half-life of the last intron is longer than the expected time to transcribe the last exon (assuming a typical elongation rate of 1.5 kb per minute). Thus, splicing is likely to impact total time for mRNA production in most genes and is likely to be rate limiting for many genes. For some genes 3’ end formation may be rate limiting. We now explicitly acknowledge and discuss these possibilities in paragraph five of the Discussion section.

4) The authors note that the first introns are often more slowly spliced than subsequent introns. What about last introns and, in particular, exon defined last introns that may rely on different mechanisms?

Last introns, regardless of splicing definition mode, do not have significantly different splicing half-lives than internal introns (Mann-Whitney U test, P = 0.062). This observation is now reported in subsection “First introns are slowly spliced and may influence rates of downstream introns”.

Reviewer #2:[…] One technical detail could be made a bit more clear. The Materials and methods section succeeds very well in describing the way that splicing kinetics are learned from the relative abundances of spliced and unspliced introns in transcripts of different lengths. The contrast of several methods for quantitative analysis is good, and makes the logic clear. But the way that the data provide inference of the distinction between the exon-defined and intron-defined splicing could be more clearly spelled out. It appears to be based solely on RIME value, and while this is a sensible starting point, one would like to know if there is any other confirmation of these calls. In other words, one would like to know error rates for this inference.

Exon and intron definition remain incompletely understood, making this topic interesting to study. Unlike nonsense-mediated mRNA decay where a specific stop codon-exon junction distance predicts decay, no hard and fast rules are accepted to predict which introns are intron-defined. However, to our knowledge experiments manipulating exon and intron lengths have always observed that shrinking of introns and expansion of exons tends to push splicing toward intron definition (never toward exon definition), and conversely for intron expansion/exon shrinking (reviewed in the Introduction section). Therefore, although we cannot ascribe a precise error rate to our classifications, both intron definition and exon definition are known to occur in *Drosophila*, and there is every reason to expect that the tendency toward exon definition increases monotonically with RIME value.

An overall assessment of goodness-of-fit of the decay model to data would be good to see.

We have now included the distribution of the residual sum of squares (RSS) of observed versus fitted splicing junction read ratios across all the introns as Figure 1—figure supplement 3. Details of RSS calculation are now included in the Materials and methods section.

Given the unusually slow kinetics of splicing of the first intron (which often hold enhancer elements), it does appear to be sensible to drop the first intron in the analysis of the factors that determine splice mode.Arguments that the splicing modes are a product of natural selection are speculative and are not based on either comparative (interspecific) analysis or on population genetic analysis. The former seems more likely to be feasible, but would entail a radical increase in the work (and so is not recommended here!). And the latter cannot be done without a battery of S2-like cells from different lines, which don't exist. Probably the relatively low frequency of polymorphism in splice sites would erode the power of any population genetic approach. So the suggestion is to make clear in the wording about the claims of selection that these are speculative inferences not based on specific evidence of past action of selection. To this reviewer, this lack does not erode the value of the paper, and the speculative remarks about natural selection do help the reader understand the nature of the question and the meaning of the results.The splicing dichotomy poses an evolutionary puzzle that might warrant some discussion. Namely, the data appear to support the idea that the splicing modes driven by the relative lengths of introns and flanking exons. But natural selection in turn can also adjust these parameters. So, is there any adaptive argument to be made why Drosophila are balanced between the use of both modes?

We agree that our discussion of how selective pressures could be influencing splicing rates is speculative (but helpful in understanding the questions and results). We have modified the text in paragraph two of the Discussion section to make this clearer and qualified our discussion of the adaptive advantages that exon definition might provide.

It is somewhat of a limitation that the entire study was done in S2 cells. Is there any hope of getting 4sU into whole organisms to get at tissue-specific characteristics?

Performing this study at an organismal level would certainly be interesting, but does not appear feasible with current technologies for metabolic labeling. Though incubation with 4sU for short time periods has no effect on gene expression or cellular function, prolonged labeling with 4sU (along with other labeled nucleotides; >24hr) has been observed to have negative effects on cell growth (Tani and Akimitsu, 2012). Extended incubation with 4sU has also been seen to inhibit rRNA synthesis in a concentration dependent manner (Burger et al., 2013). Since rRNAs have half-lives that are well beyond the scope of our labeling time course, 4sU effects on rRNA synthesis should not influence observations in this study.

There are a few issues that are not addressed and might be of interest. Drosophila make great use of alternatively spliced exons and yet alternative splicing gets very short mention in the paper. Similarly nested genes would be expected to have odd splicing dynamics, and it might be a good sanity check to see this. I expected to see more discussion of the super short introns (<50 bp) found in Drosophila – are they all intron-defined and super-rapidly spliced? Is it necessary to invoke different splicing machinery for these?

While alternative splicing is not extensively analyzed in this manuscript, we did report that introns flanking alternatively skipped exons are spliced more slowly than constitutive introns (Figure 2). A main reason for the lighter analysis of this topic is that measuring the splicing rates of alternative introns presents additional modeling challenges with regards assigning intron-exon junction reads. Our modeling of splicing half-lives assumes that intronic reads decay to zero over time, an assumption which is violated for introns alternatively included in S2 cells (our analysis in Figure 2 included introns that are alternatively included in other fly tissues but completely spliced in S2 cells). Instead of continuous labeling with 4sU, a pulse-labeling design would better allow one to both calculate the proportion of intron retention or skipped exon inclusion at each time point and to model the assignment of reads to alternative isoforms, an effort which is beyond the scope of the current study. The reviewer is correct that all of the ultra-short introns expressed in *Drosophila* S2 cells are intron-defined. However, they are quite slowly spliced as shown in Figure 2, likely because the branch points are located suboptimally relative to the 3’ splice sites (too close), leaving little or no space for the polypyrimidine track, as discussed in subsection “Introns 60-70 nt in length are spliced rapidly”.

Reviewer #3:Pai and co-workers investigated genome-wide splicing rates in Drosophila S2 cells and how these rates relate to different modes of splicing (i.e. intron and exon definition). For this, the authors employed metabolic labeling coupled to RNA sequencing, combined with elegant mathematical modeling. The study reports several interesting results, including a local maximum for splicing rates for 60-70 nt long introns (the most common length in Drosophila) and the unexpected finding that exons surrounded by very long introns are spliced the fastest and most accurately.Overall, I enjoyed the manuscript and I do not have any major concern. The following are specific comments/suggestions:1) The fact that introns with similar rates tend to more often co-occur within the same genes seems to make sense. However, I wonder whether this could be driven, at least in part, by the different elongation rates among genes. If I understood it correctly, the model they developed assumes a single elongation rate for all genes. Therefore, it is possible that some gene-level biases are introduced because of this. While this should not affect most of the conclusions in the study, the bias may be strong enough as to create patterns of significant co-occurrence within genes (all introns within a gene will have the same bias and this will be different to most other introns). This is of course difficult to test. Perhaps the authors could use their data, consisting of three time points, to roughly estimate elongation rates and group genes based on these. Irrespectively, I may be a good idea to add a brief note about it.

As discussed above in our response to Reviewer #1’s first comment, our data do not provide a basis for reliable estimation of transcription elongation rates, but our simulations indicate that variation in rates within the range of 0.5 kb/min to 4 kb/min have only a modest effect on splicing half-lives (Figure 1—figure supplement 3). However, we now explicitly acknowledge that different transcriptional elongation rates, and especially non-uniform rates across a gene, might have an effect on absolute and relative measurements of splicing rates in subsection “Introns 60-70 nt in length are spliced rapidly” of the manuscript. As the reviewer points out, our assumption of a single elongation rate for all genes and across introns in the same gene could create biases that lead to more consistent splicing rates across introns within a gene. To estimate the extent to which our observations of gene-specific splicing rate differences might be affected by this assumption, we randomly sampled a transcription rate between 0.5 kb/min and 4 kb/min independently for each intron and re-analyzed consistency in splicing rates between introns from the same gene. We find that the variance across splicing half-lives of introns from the same gene is still significantly lower than randomly sampled introns, even when varying the transcription rates across introns in the same gene. We have now included this control analysis as Figure 4—figure supplement 1.

2) Figure 4: given the results in Figure 4, it would be good to also have a comparison of half-life SDs for randomly sampled introns within groups of similar lengths, to see if this is the main feature driving the signal (rather than gene co-occurrence).

The randomly sampled introns in Figure 4 were chosen such that each simulated gene unit has the same distribution of intron lengths as a corresponding real gene using an intron length binning approach. Thus, we have controlled for effects of intron (and gene) length on variation in intron half-lives across genes.

3) Subsection “Exon definition is associated with faster and more accurate splicing” and Figure 3: this plot and the associated description in the text are not very easy to understand. The reference to stripes is ambiguous. I do not have a better suggestion on how to represent the data, but at least the wording in the text could be improved. It may help to label/highlight the sections of interest in the plot.

We have now added a supplementary figure that delineates the particular regions of interest that are highlighted in the text (Figure 3—figure supplement 1). We have also clarified the terminology that we use to describe the figure in the manuscript.